# Cerebellar re-encoding of self-generated head movements

**Guillaume P Dugué[1†], Matthieu Tihy[1†], Boris Gourévitch[2], Clément Léna[1*]**

[1]Neurophysiology of Brain Circuits Team, Institut de Biologie de l'École Normale Supérieure, Inserm U1024, CNRS UMR8197, École Normale Supérieure, PSL Research University, Paris, France; [2]Genetics and Physiology of Hearing Laboratory, Inserm UMR1120, University Paris 6, Institut Pasteur, Paris, France

**Abstract** Head movements are primarily sensed in a reference frame tied to the head, yet they are used to calculate self-orientation relative to the world. This requires to re-encode head kinematic signals into a reference frame anchored to earth-centered landmarks such as gravity, through computations whose neuronal substrate remains to be determined. Here, we studied the encoding of self-generated head movements in the rat caudal cerebellar vermis, an area essential for graviceptive functions. We found that, contrarily to peripheral vestibular inputs, most Purkinje cells exhibited a mixed sensitivity to head rotational and gravitational information and were differentially modulated by active and passive movements. In a subpopulation of cells, this mixed sensitivity underlay a tuning to rotations about an axis defined relative to gravity. Therefore, we show that the caudal vermis hosts a re-encoded, gravitationally polarized representation of self-generated head kinematics in freely moving rats.

**\*For correspondence:** lena@ biologie.ens.fr

[†]These authors contributed equally to this work

**Competing interests:** The authors declare that no competing interests exist.

## Introduction

Self-orientation is largely dependent on modalities that document head movements, including neck proprioception, optic flow and, most notably, vestibular inputs (*Angelaki and Cullen, 2008*). Vestibular signals are essential for stabilizing gaze (*du Lac et al., 1995*) and for computing head direction, spatial maps and navigation trajectories (*Stackman et al., 2005*; *Yoder and Taube, 2014*; *Wallace et al., 2002*; *Klier and Angelaki, 2008*; *Rochefort et al., 2011*; *Rochefort et al., 2013*; *Rondi-Reig et al., 2014*). These signals originate from two categories of skull-anchored inertial sensors: gyroscope-like structures (semi-circular canals), which transduce head angular velocity, and accelerometer-like structures (otolith organs), which are activated indifferently by accelerated linear motion and by gravity. Gravity provides an absolute directional cue on the external world and is effectively derived from vestibular inputs in the vestibular system (*Merfeld et al., 1999*; *Angelaki et al., 1999*), allowing the brain to align the axes of eye rotations with the direction of gravity (*Hess and Angelaki, 1997*; *Hess and Angelaki, 2003*). Head direction cells are also anchored to a reference frame aligned with gravity rather than to the animal's locomotor plane (*Taube et al., 2013*; *Finkelstein et al., 2016*; *Wilson et al., 2016*; *Olson et al., 2017*); their activity, which relies on the temporal integration of head angular velocity signals (*Song and Wang, 2005*), thus also requires information on head orientation relative to gravity (head tilt) (*Yoder and Taube, 2009*). Indeed, identical activations of semi-circular canals may affect the azimuth and elevation of the head in very different ways depending on head tilt: for example, a rotation about the interaural axis will lead opposite changes of azimuth if the head is tilted with the left or right ear down. Therefore, understanding how the brain computes head direction requires to identify the neuronal substrate of the operations transforming skull-bound angular velocity into changes of azimuth and elevation.

Lesion data suggest that the caudal cerebellar vermis, a brain region receiving multimodal sensory cues related to head kinematics and orientation (*Quy et al., 2011*; *Yakusheva et al., 2013*), plays a pivotal role in the discrimination of gravity (e.g. *Kim et al., 2015*; *Tarnutzer et al., 2015*; *Lee et al., 2017*). Moreover, a distinct population of caudal cerebellar Purkinje cells in monkeys dynamically reports head tilt during passive whole-body movements (*Yakusheva et al., 2007*; *Laurens et al., 2013a*; *Laurens et al., 2013b*). We therefore hypothesized that this structure might also host a representation of head rotations anchored to the direction of gravity.

A considerable literature has described the responses of caudal vermis Purkinje cells to passively experienced head movements (reviewed in *Barmack and Yakhnitsa, 2011*). However, these movements only covered the lower range of frequencies and amplitudes observed during active self-motion (*Carriot et al., 2014*; *Carriot et al., 2015*; *Pasquet et al., 2016*; *Carriot et al., 2017*). Moreover, despite the remarkable linearity of early vestibular information processing (*Bagnall et al., 2008*; but see *Massot et al., 2011* and *Sadeghi et al., 2007*), the high amplitude of active movements might recruit vestibular afferents in a non-linear way (*Hullar et al., 2005*; *Schneider et al., 2015*). In addition, studies in mice and monkeys have revealed that active and passive head movements are processed in fundamentally different ways within the vestibular nuclei (*McCrea et al., 1999*; *Cullen and Roy, 2004*; *Roy and Cullen, 2004*), which are highly interconnected with the caudal cerebellar vermis. Thus, the principles of vestibular coding in passive conditions might not apply to the active condition. We therefore decided to study the encoding of head movements in the caudal cerebellar vermis in freely moving rats, while monitoring the movements of their head using a miniature inertial sensor.

## Results

### Kinematics of self-generated head movements

Combined recordings of cerebellar activity and head movements were obtained in 16 freely moving rats (*Figure 1A*). Spontaneous exploratory behavior produced a wide variety of head positions and movements. Our inertial device captured the same parameters as vestibular organs: rotations and accelerations in a head-bound reference frame. In average, head rotations occurred more frequently and swiftly along the pitch and yaw axes than along the roll axis (*Figure 1—figure supplement 1A, B*), with typical angular speed in the 18–287 °/s range (average 2.5–97.5% percentiles calculated for velocities >15 °/s, n = 16 rats). Angular velocity ($\Omega$) signals displayed multi-peaked power spectra spanning frequencies up to 20 Hz (*Figure 1—figure supplement 1C*) and showed strong temporal autocorrelation over a short timescale (<0.2 s, *Figure 1—figure supplement 2A*). The acceleration signal (A) was composed of a gravitational ($A^G$) and a non-gravitational ($A^{nG}$) component. The gravitational component could be described as a vector $a^G$ with a constant norm (1 $g$) and a fixed orientation in the earth reference frame, but whose coordinates varied in the sensor (head-bound) reference frame during changes of head orientation relative to gravity (head tilt). The direction of $a^G$ in the sensor frame thus reflected head tilt (see *Video 1*). We separated the two components of acceleration using an orientation filter algorithm (*Figure 1—figure supplement 1D*; see Appendix and *Madgwick et al., 2011*). $A^G$ accounted for almost all (99%) of the power of the acceleration below 2 Hz, and for only 9% of it in the 2–20 Hz range (n = 16 rats, *Figure 1—figure supplement 1E*), indicating that the low-frequency component of acceleration (<2 Hz) mostly contained head tilt information. Consistent with this, $A^G$ displayed temporal autocorrelation over long timescales (<5 s, *Figure 1—figure supplement 2B*). $A^{nG}$ varied at the same timescale as $\Omega$ and exhibited the same order of magnitude, and temporal correlation pattern with $\Omega$, as linear tangential acceleration predicted from head rotations (see *Figure 1—figure supplement 2C–E* and Appendix). This suggests that, in our conditions, $A^{nG}$ signals arose primarily from head rotations.

### Caudal cerebellar units exhibit a mixed sensitivity to head angular velocity and gravitational acceleration

A total of 86 units were recorded (*Figure 1A,B* and *Figure 1—figure supplement 3A–C*) and classified into putative Purkinje cells (90%), Golgi cells (5%) and mossy fibers (5%) using established criteria (*Van Dijck et al., 2013*, see *Figure 1—figure supplement 3E* and Appendix). Putative Purkinje cells exhibited irregular inter-spike intervals (ISI) at rest (average CV: 0.95 ± 0.58, calculated for

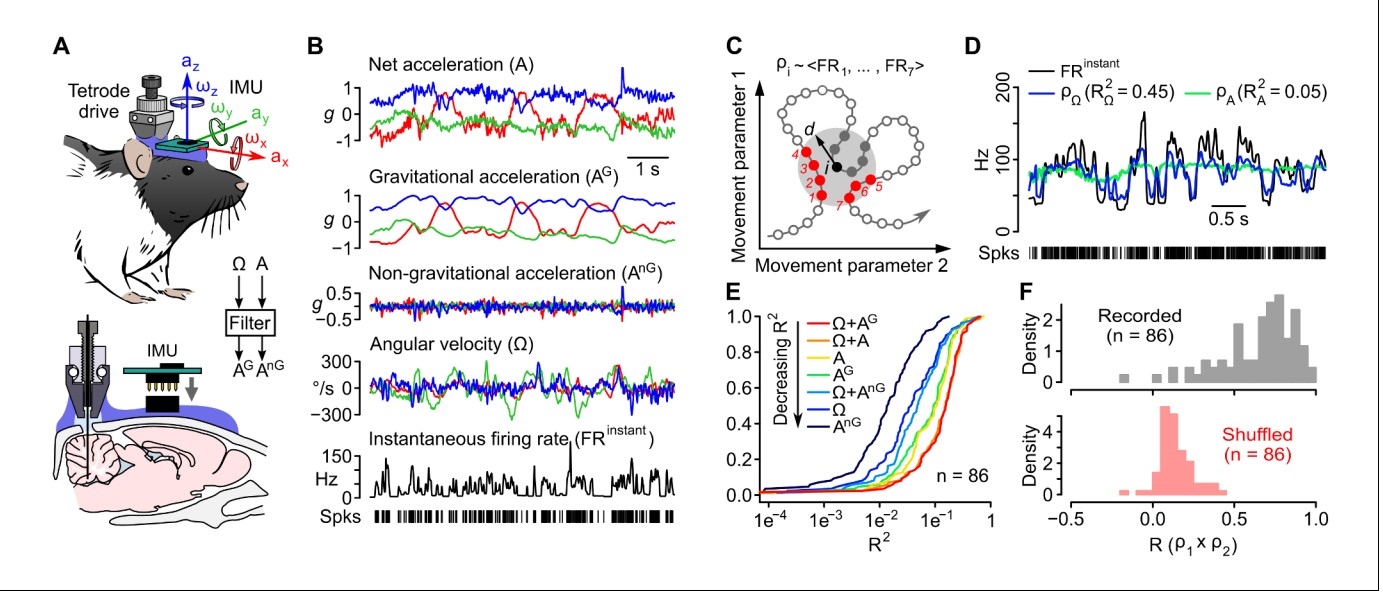

**Figure 1.** Caudal cerebellar units are sensitive to different combinations of rotational and gravitational information. (A) Orientation of the inertial measurement unit (IMU) on the animal's head and tetrode placement. An algorithm ('filter', see Appendix) was used to calculate the gravitational ($A^G$) and non-gravitational ($A^{nG}$) components of acceleration (A) using angular velocity ($\Omega$) information. (B) Traces showing the instantaneous firing rate ($FR^{instant}$) of an example unit and inertial signals recorded simultaneously (A and $\Omega$) or calculated offline ($A^G$ and $A^{nG}$). (C) Principle of the model-free resampling method (see Appendix for details). Recordings of head movements can be described as sequences of points in a multidimensional parameter space (circles and line, here represented in a 2D space). At a given time point $i$ (black circle), the estimated firing rate $\rho_i$ is the mean of $FR^{instant}$ values observed for neighboring points in the parameter space within a distance $d$ (red circles) that did not occur immediately before or after $i$ (filled gray circles). (D) Firing rate estimates calculated using $\Omega$ ($\rho_\Omega$) or A ($\rho_A$) and $FR^{instant}$ of an example unit. The values of the square of the Pearson correlation coefficient ($R^2$) between $FR^{instant}$ and firing rate estimates are indicated under parenthesis. (E) Cumulative distribution of $R^2$ for firing rate estimates calculated using different combinations of inertial parameters (n = 86 units). (F) Distribution of Pearson correlation coefficients ($R$) between independent firing rate estimates, calculated using the combination of inertial parameters yielding the best estimate for each unit (gray histogram, mean $R = 0.65 \pm 0.23$, n = 86 units). A null distribution was calculated using shuffled spike trains (red histogram, average from 10 iterations, mean $R = 0.14 \pm 0.10$, n = 86 units, see Appendix).

The following figure supplements are available for figure 1:

**Figure supplement 1.** Head angular velocity and acceleration signals in freely moving rats.

**Figure supplement 2.** Geometrical and temporal coupling of head inertial signals during self-motion.

**Figure supplement 3.** Isolation and classification of recorded units.

periods of immobility isolated from 76 cells), resulting in sharp fluctuations of the instantaneous firing rate even during immobility (*Figure 1—figure supplement 3D*).

We first examined to which extent these firing rate fluctuations could be explained by head movements. Simple linear models may be inadequate for describing nonlinear regimes of responses of vestibular afferents observed during naturalistic head movements (*Schneider et al., 2015*) or for capturing complex receptive fields reflecting a nonlinear remapping of head kinematics into a gravitationally polarized reference frame (*Green and Angelaki, 2007*). We therefore designed a model-free approach based on a jackknife resampling technique (see Appendix), which makes no assumption on the nature of the link between inertial parameters and firing rate, except that similar inertial configurations yield similar firing rate. At each time point of a recording, this method identifies the corresponding values of the inertial parameters (e.g. $\Omega$ alone, or $\Omega$ + A, etc.), and computes the average of instantaneous firing rates observed at other time points with similar values of inertial signals (with an adjustable time delay between the firing rate and inertial parameters, see Appendix and *Figure 1C*). The resulting time series represents an estimate of firing rate modulations that can

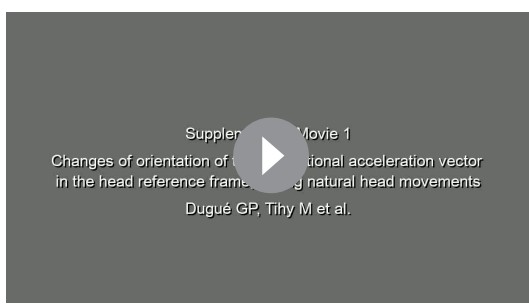

**Video 1.** Changes of orientation of the gravitational acceleration vector ($a^G$) in the head reference frame during natural head movements. This movie illustrates how gravitational acceleration information can be used to track head orientation relative to gravity (head tilt). Left: 3D view of a rat skull animated using a sequence of head rotations acquired from a freely moving rat. The head was positioned in 3D using the output of the orientation filter. Because the filter only outputs head tilt information, and not absolute orientation (comprising azimuthal information), the naso-occipital ($x$) axis of the head was maintained aligned in the same vertical plane. $a^G$ is represented by a purple arrow. Right: trajectory of $a^G$ (purple arrow) corresponding to the same movements and represented in the head reference frame. Because $a^G$ has a constant norm (of 1 $g$), its trajectory is contained within a sphere (represented in gray).

be attributed to the influence of this set of parameters and is equivalent to the global output expected from a population of cells that would share the corresponding sensitivity profile. The square of the Pearson correlation coefficient ($R^2$) between estimated and observed firing rates was taken as a measure of the fraction of instantaneous firing rate fluctuations that could be explained by a particular set of inertial parameters (firing rate predictability, *Figure 1D*).

Overall, we found that firing rate predictability was greater when considering $\Omega$ combined with either A or $A^G$ (mean $R^2 = 0.17 \pm 0.13$ in both cases, n = 86 units; *Figure 1E*). Considered alone, all inertial parameters ($\Omega$, A, $A^G$ or $A^{nG}$) explained significantly smaller fractions of firing rate fluctuations than combinations of $\Omega$ and A or $A^G$ (p<0.005, n = 86 units). $R^2$ values were always greater for $A^G$-based than for $A^{nG}$-based estimates (p<8e-9 for both $A^G$ vs. $A^{nG}$ and $\Omega + A^G$ vs. $\Omega + A^{nG}$, n = 86 units), showing that gravitational information dominated the effect of acceleration on firing rate. $R^2$ values for firing rate estimates obtained with $\Omega$ or $\Omega + A^{nG}$ were similar (p=0.20, n = 86 units), consistent with a redundancy of these parameters due to their coupling (*Figure 1—figure supplement 2E*). This analysis suggests that cerebellar units preferentially exhibited a mixed sensitivity to head rotations and head tilt.

To assess the robustness of our method, we examined the correlation between independent firing rate estimates computed using non-overlapping (alternating) portions of the same recordings (see Appendix). The Pearson correlation coefficients ($R$) between independent estimates were mostly above 0.5 (mean $R = 0.65 \pm 0.23$, n = 86, top panel of *Figure 1F*) while the null distribution computed using shuffled spike trains exhibited significantly smaller $R$ values centered near zero (mean $R = 0.14 \pm 0.10$, n = 86 units; p=$2.2 \times 10^{-16}$, bottom panel of *Figure 1F*) showing the ability of our method to consistently capture the link between head movements and firing rate modulations.

## The influence of head movements on firing rate is shared by neighboring units and is independent of visual cues, but varies when movements are self-generated or passively experienced

The cerebellar cortex is divided into narrow functional zones, the microzones (*Apps and Hawkes, 2009*; *Dean et al., 2010*), to which neighboring units (recorded simultaneously by a tetrode) likely belong. The instantaneous firing rate of neighboring units (*Figure 2A*) indeed displayed positive correlations ($R = 0.14 \pm 0.26$, n = 35 pairs, p=0.0014, one-sampled Wilcoxon test) that were lost if the spike train of one unit was time-reversed ($R = 0.00 \pm 0.03$, p=0.49, one-sampled Wilcoxon test, *Figure 2C*). To test whether these correlations were due to a similar dependency on inertial parameters (versus a shared, movement-independent entrainment of neighboring units), we isolated the movement-dependent part of the firing rate with our resampling method (*Figure 2B*) and examined their correlations. Correlations were higher (p=0.0074, paired Wilcoxon test) when comparing firing rate estimates ($R = 0.31 \pm 0.47$, p=0.0013, one-sampled Wilcoxon test, *Figure 2C*) than when comparing instantaneous firing rates (*Figure 2D*), showing that neighboring unit tended to share similar sensitivities to head movements.

We then examined whether changing experimental conditions affected the units' sensitivity by comparing independent firing rate estimates obtained by resampling from the same or from

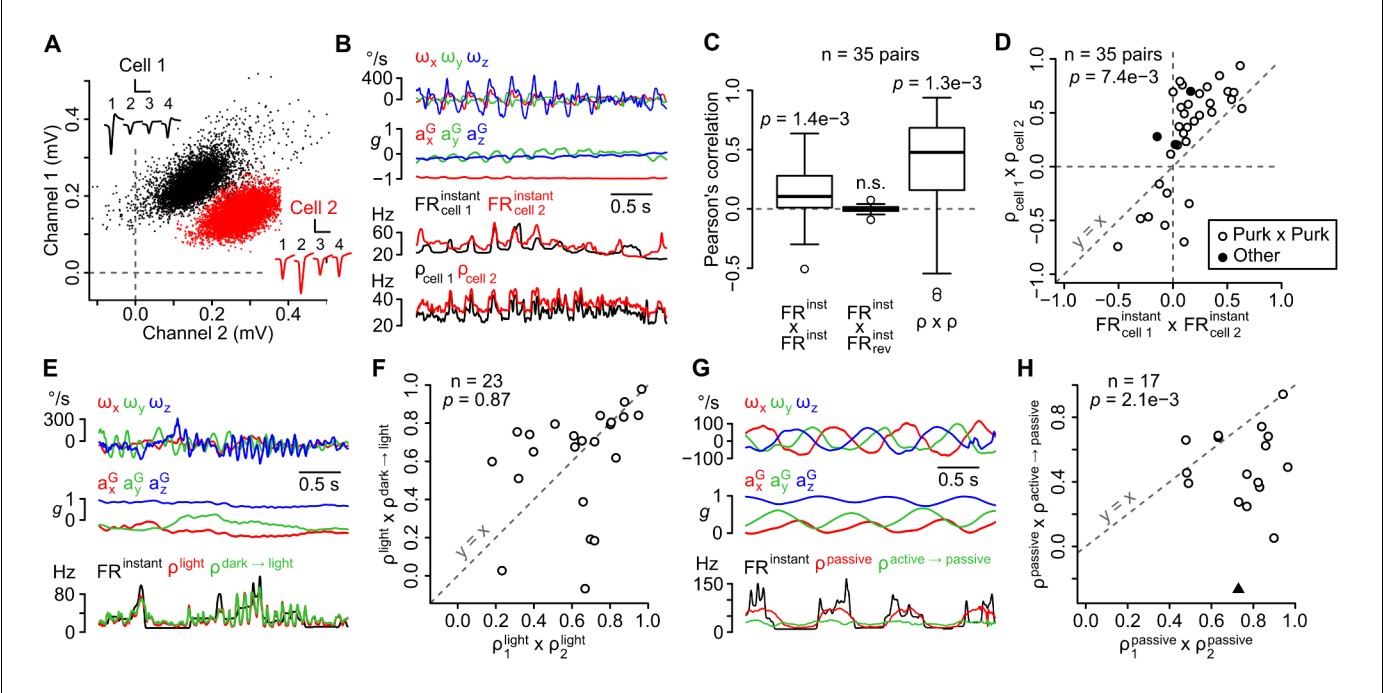

**Figure 2.** The sensitivity of recorded units is similar in the same recording site, does not depend on visual cues, but differs between active and passive movements. (A) Amplitude of sorted spikes on a pair of channels and average spike waveforms of two neighboring units (scale bars: 0.3 mV and 1 ms for cell 1, 0.15 mV and 1 ms for cell 2). Channel numbers are indicated above the waveforms. (B) Example traces showing inertial parameters and the instantaneous and estimated firing rates of the two units shown in A. (C) Boxplots of Pearson correlation coefficients between instantaneous ($FR^{instant} \times FR^{instant}$) or estimated ($\rho \times \rho$) firing rates of neighboring units (n = 35 pairs). The correlation between $FR^{instant}$ was lost if the firing rate of one unit was time-reversed ($FR^{instant} \times FR_{rev}^{instant}$, p=0.49; p-values above the boxplots were computed using a one-sample Wilcoxon test – null hypothesis: median = 0). (D) Graph comparing Pearson correlation coefficients between firing rate estimates ($\rho_{cell\ 1} \times \rho_{cell\ 2}$) and instantaneous firing rates ($FR_{cell\ 1}^{instant} \times FR_{cell\ 2}^{instant}$) of neighboring units (p=0.0074, paired Wilcoxon test, n = 35 pairs). Pairs of putative Purkinje cells (n = 32) are shown in white. The diagonal dashed line represents the identity line. (E) Example traces showing inertial parameters and $FR^{instant}$ for one example unit recorded in the light block. Color traces are firing rate estimates for the same recording calculated using data from the same block ($\rho^{light}$) or from the dark block ($\rho^{dark \rightarrow light}$). (F) Graph comparing Pearson correlation coefficients between independent firing rate estimates in the light block ($\rho_1^{light} \times \rho_2^{light}$) and between estimates of the firing rate in the light block calculated using data from either the light or dark block ($\rho^{light} \times \rho^{dark \rightarrow light}$). The p-value was computed using a paired Wilcoxon test. All units corresponded to putative Purkinje cells (n = 23). The diagonal dashed line represents the identity line. (G) Example traces showing inertial parameters and $FR^{instant}$ for one example unit recorded in the passive block. Color traces are firing rate estimates for the same recording calculated using data from the same block ($\rho^{passive}$) or from the passive block ($\rho^{active \rightarrow passive}$). (H) Graph comparing Pearson correlation coefficients between independent firing rate estimates in the passive block ($\rho_1^{passive} \times \rho_2^{passive}$) and between estimates of the firing rate in the passive block calculated using data from either the passive or active block ($\rho^{passive} \times \rho^{active \rightarrow passive}$). The p-value was computed using a paired Wilcoxon test. All units corresponded to putative Purkinje cells, except one classified as a putative Golgi cell (black triangle, n = 17 units in total). The diagonal dashed line represents the identity line.

The following figure supplement is available for figure 2:

**Figure supplement 1.** Statistics of head kinematics during active and passive movements.

different conditions (see Appendix); if changing the condition has little effect on the firing, these estimates should exhibit similar degrees of correlation. Overall, the presence or absence of light did not change the correlation between independent estimates (p=0.87, paired Wilcoxon test, n = 23, *Figure 2E,F*), showing a limited influence of visual cues on the units' sensitivity to head movements. We also tested whether passively-applied whole-body movements (which neither recruit proprioceptive inputs nor produce a motor efferent copy) drove the units similarly compared to the active situation. These passive movements explored a subset of the inertial configurations observed in the active condition (*Figure 2—figure supplement 1*); we could therefore use our resampling method to estimate the firing rate in the passive session using observations from the active session. The resulting estimates significantly differed from estimates obtained using observations from the

passive session (p=0.0021, paired Wilcoxon test, n = 17, *Figure 2G,H*), suggesting that in many cases the units' coding schemes differed when movements were self-generated or passively experienced.

## Subsets of caudal cerebellar units are specifically tuned to either rotational or gravitational information

According to our model-free approach, most units exhibited a mixed gravitational and rotational sensitivity (*Figure 1E*), but a fraction of them appeared to be mostly tuned to $\Omega$ or $A^G$. We then first examined the nature of the link between the firing rate and these inertial parameters in these cells. We isolated 6 $\Omega$-selective and 12 $A^G$-selective units by picking cases for which the firing rate predictability was at least eightfold greater for one parameter than the other (*Figure 3A*).

The sensitivity of $\Omega$-units was examined by computing 'inertio-temporal receptive fields' (average instantaneous firing rate timecourse around specific angular velocity values at lag 0; see Appendix). These plots showed a bidirectional modulation of the firing rate by specific combinations of rotations, since the firing rate was increased or decreased at a lag close to 0, that is, close to the occurrence of specific rotation values (*Figure 3B*). 3D plots representing the firing rate as a function of the three components of $\Omega$ (calculated for the lag showing the strongest modulation of firing rate) revealed clear firing rate gradients along specific directions of rotation, showing that these units were tuned to specific 3D rotations (*Figure 3C* and *Video 2*).

To quantify the sensitivity of $\Omega$-units to angular velocity, we used a simple regression model in which the instantaneous firing rate is described as a combination of the roll, pitch and yaw velocities. The coefficients of the fit define the coordinates of a rotation sensitivity vector, whose norm, in Hz/($^\circ$/s) or $^{\circ-1}$, represents the gain of the unit's response, and whose direction represents the unit's preferred rotation (*Figure 3D*). For each unit, the linear fit was calculated with a variable time delay (lag) between the firing rate and angular velocity; the presence of a peak in the sensitivity vs. lag curve (see *Figure 3E* for average gain vs. lag curves) allowed us to identify an optimal lag at which the unit's gain was maximal. The sensitivity vector calculated at the optimal lag was defined as the unit's optimal rotational sensitivity vector ($\boldsymbol{\omega_{opt}}$). The gain calculated at the optimal lag was greater for $\Omega$-units (0.28 $\pm$ 0.15 $^{\circ-1}$, n = 6) than for $A^G$-units and other units with mixed sensitivity (0.03 $\pm$ 0.02 $^{\circ-1}$, n = 7, p=0.0012 and 0.07 $\pm$ 0.05 $^{\circ-1}$, n = 53, p=0.00022, respectively; calculated only for units with a significant $\boldsymbol{\omega_{opt}}$ vector; see Appendix). The optimal lag of $\Omega$-units was centered around zero (0.5 $\pm$ 21.8 ms, n = 6) while the one of mixed units was greater (29.8 $\pm$ 142.4 ms, n = 53), although not significantly (p=0.072). $\Omega$-units corresponded to putative mossy fibers (n = 2) and putative Purkinje cells (n = 4), and exhibited preferred rotation axes clustered around the excitatory direction of semi-circular canals (*Figure 3F*). Units with mixed sensitivity corresponded in majority (96%) to putative Purkinje cells.

The tuning of $A^G$-units was examined by computing their average firing rate as a function of head tilt; head tilt was defined as the orientation of the gravity vector $\boldsymbol{a^G}$ in head coordinates (*Video 1*), which could be mapped on a sphere (*Figure 3G*) and then in two dimensions using an equal-area projection (*Figure 3H*). The resulting plots confirmed that $A^G$-units were strongly modulated by head tilt with simple receptive fields (region of increased firing rate; *Figure 3H*), contrarily to $\Omega$-units which were not modulated by head tilt (the CV of firing rate across head tilts was higher for $A^G$-units than for $\Omega$-units: 2.28 vs. 0.29, p=0.0047, *Figure 3I*). $A^G$-units were classified as putative mossy fibers (n = 1), Golgi cells (n = 3) and Purkinje cells (n = 8). Overall, these data show that a fraction (20%) of caudal cerebellar units displayed selective tuning to either head angular velocity or head tilt; most of our putative granular layer units (6/8) belonged to these categories.

## The rotational sensitivity of most caudal cerebellar units is tilt-dependent

As shown above, most units displayed a mixed sensitivity to rotational and gravitational information and were not classified as $\Omega$-units or $A^G$-units. A direct examination of inertio-temporal receptive fields for different head orientations showed that the rotational sensitivity of these units was indeed often highly tilt-dependent. *Figure 4A–F* shows two example units for which the apparent sensitivity to yaw angular velocity ($\omega_z$) increased (*Figure 4A–C*) or even reversed (*Figure 4D–F*) for nose down vs. nose up head situations (i.e. positive vs. negative values of $a_x^G$). Inertio-temporal receptive fields

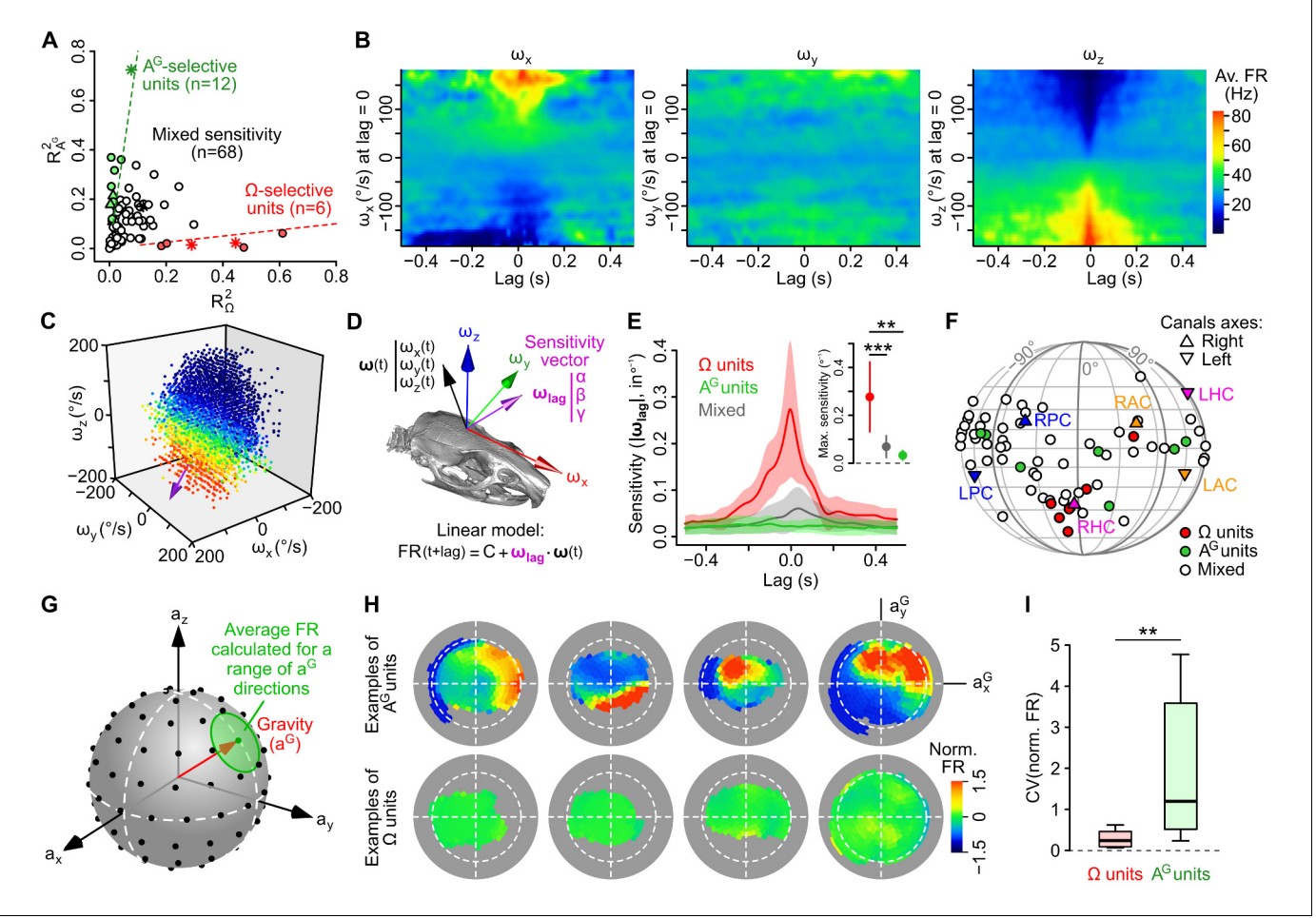

**Figure 3.** Subsets of caudal cerebellar units display preferential sensitivity to either head angular velocity or head tilt. (**A**) Comparison of $R^2$ values calculated based on gravitational acceleration ($R^2_{A^G}$) or angular velocity ($R^2_{\Omega}$) for all 86 units. Dashed lines delineate units with an $R^2$ at least eight times greater for one parameter than the other and greater than 0.1 for the preferred parameter. Putative Purkinje cells, Golgi cells and mossy fibers are represented by empty circles, filled triangles and stars, respectively. (**B**) Inertio-temporal receptive fields of one example $\Omega$-unit. (**C**) Firing rate (color-coded) of one $\Omega$-unit (same as *B*) plotted as a function of the three components of angular velocity (see also *Video 1*) at the optimal lag. The colorbar is the same as in *B*. The unit's optimal sensitivity vector at optimal lag (calculated using a linear regression, see *D*) is represented in purple (arbitrary scale). (**D**) Linear model used to characterize the units' rotational tuning. For a given lag, the model assumes a linear tuning of firing rate to a preferred sensitivity vector $\boldsymbol{\omega}_{lag}$. (**E**) Average (±SD) rotational sensitivity (norm of $\boldsymbol{\omega}_{lag}$) plotted vs. lag values for $\Omega$-units (n = 6), $A^G$-units (n = 7) and other ('mixed') units (n = 53). Note that $A^G$-units exhibit very weak rotational sensitivity. Only units with significant sensitivity were included (see Appendix). Inset: average (±SD) sensitivity of $\Omega$-, $A^G$- and mixed units at their optimal lag. **p=1.2 × 10⁻³, ***p=2.2 × 10⁻⁴. (**F**) Direction of optimal sensitivity vectors of $\Omega$-units, $A^G$-units and mixed units plotted on a pseudocylindrical projection. Triangles point up (resp. down) represent the excitatory direction of rotation of right (resp. left) semi-circular canals. LPC/RPC: left/right posterior canals; LHC/RHC: left/right horizontal canals; LAC/RAC: left/right anterior canals. (**G**) Calculation of tilt-dependent rate maps (see Appendix). The average firing rate was calculated for directions of the gravity vector ($a^G$, in head coordinates) falling within 20° (green circle) of a series of points evenly distributed over a sphere (black dots). (**H**) Lambert azimuthal equal-area projections of spherical tilt-dependent rate maps for four example $A^G$-units (top) and four example $\Omega$-units (bottom). Dashed circles represent the equator (90° head tilt). (**I**) Boxplot of the CV of firing rate values in tilt-dependent rate maps for $\Omega$-units (n = 6) and $A^G$-units (n = 12). **p=0.0047.

(*Figure 4A,D*) clearly showed changes in the modulation of the cells by rotation as a function of lag between the nose up and nose down situations; this is clearly seen as a change of the sensitivity of the cells to $\omega_z$ (i.e. slope of the firing rate vs. $\omega_z$ linear regression) for different time lags (*Figure 4B, E*). At the optimal lag (that maximizes the sensitivity for each cell), the modulation of firing rate by angular velocity progressively changed as a function of the head elevation angle (*Figure 4C,F*). The occurrence of such head tilt-dependent modulation of rotational sensitivity was quantified by a stability index ($\sigma$) quantifying the effect of $a_x^G$ ($\sigma_x$) or $a_y^G$ ($\sigma_y$) on the direction of rotational sensitivity

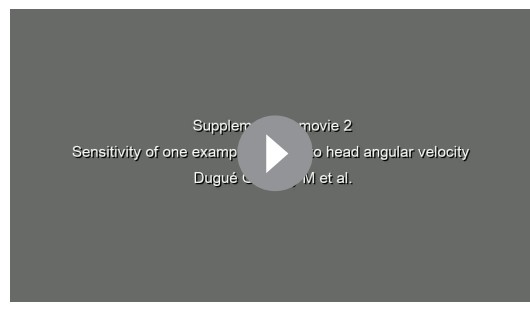

**Video 2.** 3D plot of the sensitivity of one example Ω-unit to head angular velocity (see Appendix). The unit is the same as the one shown in *Figure 3F*. The average firing rate is color coded. The optimal rotational sensitivity vector $\omega_{opt}$ calculated using a linear regression (see Appendix) is shown in purple (arbitrary scale) to show its alignment with the gradient of firing rate values.

vectors (a maximal value of 1 indicating identical directions for opposite head orientations; see Appendix). This index was lower for non-Ω-units ($\sigma_x = 0.36 \pm 0.32$ and $\sigma_y = 0.41 \pm 0.29$, n = 60) than for Ω-units ($\sigma_x = 0.73 \pm 0.13$ and $\sigma_y = 0.88 \pm 0.06$, n = 6; p=0.0027 and p=0.00011, respectively, *Figure 4G,H*), confirming that the rotational tuning of the largest fraction of our cell population was head tilt-dependent.

## A fraction of caudal cerebellar units encodes head rotations in a gravity-centered reference frame

The above data suggest that some units employed a rotational coding scheme that takes into account the direction of gravity, raising the possibility that some of them might encode head rotations in a reference frame aligned with the earth-vertical direction. To explore this, we computed rotational sensitivity vectors ($\omega_{opt}$) for all head tilt angles explored by the animal, using

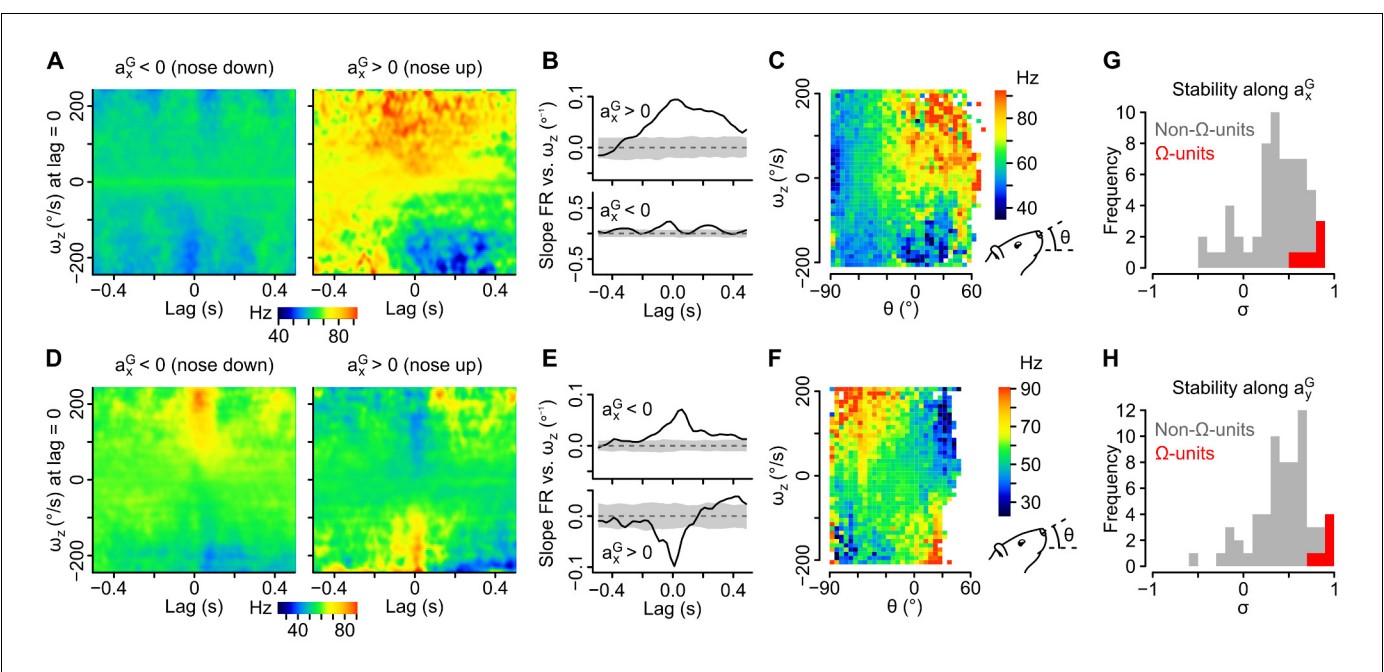

**Figure 4.** Tilt-dependence of rotational sensitivity in units with mixed gravitational and rotational sensitivity. (A–F) Example units exhibiting a pitch tilt-dependent modulation of their apparent sensitivity to yaw velocity ($\omega_z$, measured in the sensor's frame). In one unit (A–C), $\omega_z$ sensitivity is visible for nose up orientations only ($a_x^G > 0$). In the other unit (D–F), $\omega_z$ sensitivity reverses for nose up vs. nose down orientations ($a_x^G > 0$ vs. $a_x^G < 0$). (A, D) Inertio-temporal receptive fields for $\omega_z$ for nose up vs. nose down orientations. (B, E) Slope of the firing rate vs. $\omega_z$ linear regression (calculated from the receptive fields in A and D), computed for different lag values in the nose up and nose down orientations. Shaded area represent the mean slope ±2 × SD calculated using shuffled spike trains (100 iterations). (C, F) Histogram showing the average firing rate (color coded) as a function of $\omega_z$ and of the head's pitch angle (θ). (G–H) Histograms of the stability index calculated for positive vs. negative values of $a_x^G$ (G) and for positive vs. negative values of $a_y^G$ (H). The stability index was used to quantify the influence of head tilt on the direction of rotational sensitivity over a given lag range (see Appendix). Values close to 1 (resp. –1) denote a weak (resp. strong) influence of head tilt on the direction of rotational sensitivity. Histograms for non-Ω-units with significant rotational sensitivity (n = 60 units) are colored in gray and histograms for Ω-units (n = 6 units) are colored in red.

angular velocity signals expressed either in an internal (head-bound), or in an external (earth-bound), reference frame (*Figure 5B*, see Appendix). We reasoned that $\boldsymbol{\omega_{opt}}$ vectors of units encoding rotations in one reference frame (internal of external) should remain aligned (tilt-independent) only when calculated in this particular frame. As a means to visualize the effect of head tilt on these vectors, we plotted them on a sphere at the coordinates corresponding to the head tilt (i.e. the coordinates of gravity in the head reference frame, see *Figure 3G* and *Video 1*) for which they were calculated (*Figure 5A*). *Figure 5C,D* shows two example units with different tuning properties. In the left unit (*Figure 5C*), $\boldsymbol{\omega_{opt}}$ vectors appeared more consistently aligned when calculated in external (vs. internal) coordinates, while the opposite was observed for the right unit (*Figure 5D*, see *Video 3* for a 3D version of these plots). This was confirmed by examining the collinearity of $\boldsymbol{\omega_{opt}}$ vectors as a function of the angular distance between their localization on the sphere: collinearity decreased with

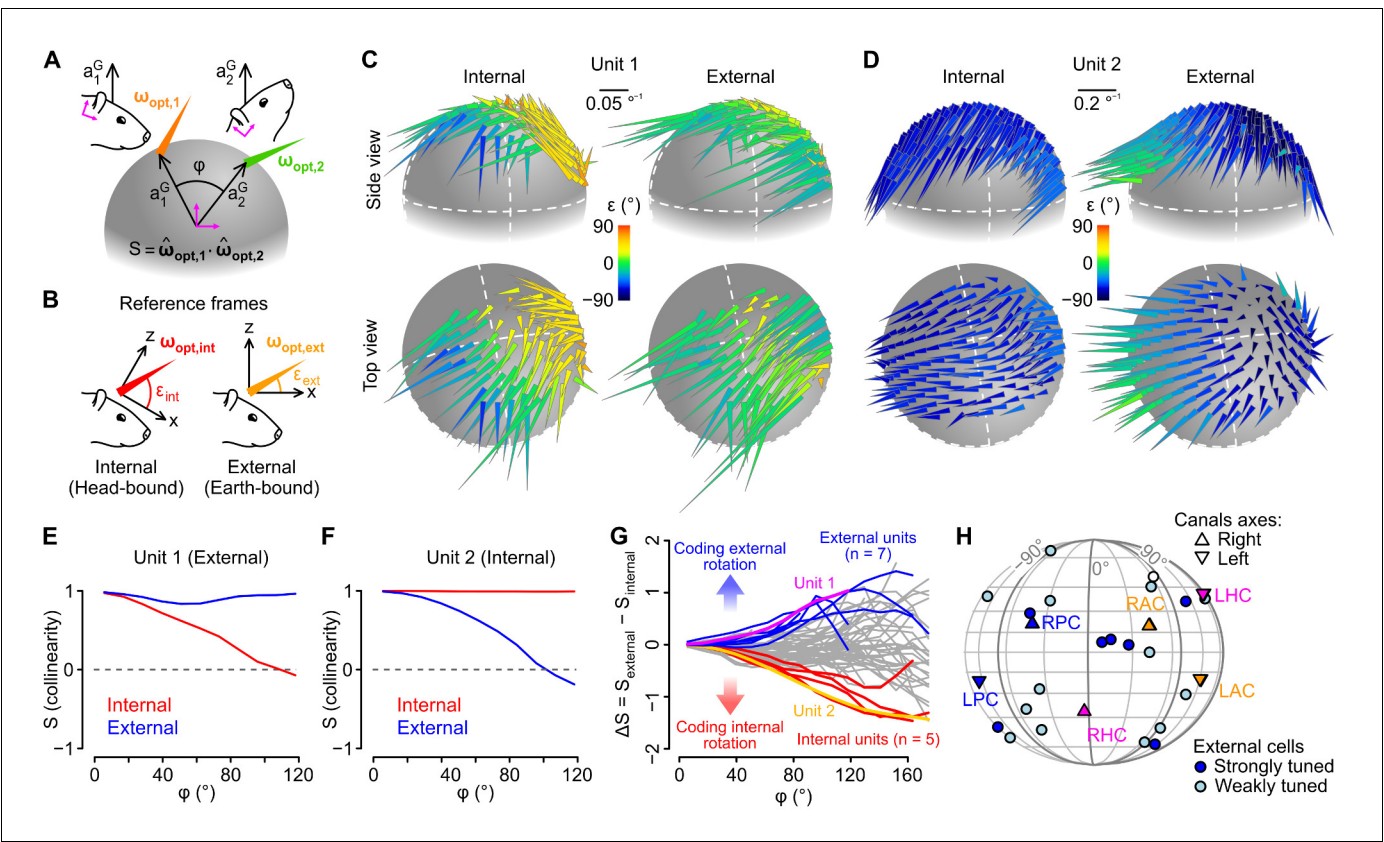

**Figure 5.** Different caudal cerebellar units encode head rotations in a head-bound or earth-bound reference frame. (A) Method used to examine the influence of head tilt on rotational sensitivity. Optimal sensitivity vectors ($\boldsymbol{\omega_{opt}}$) were calculated for different orientations of the gravity vector ($\boldsymbol{a^G}$) in head coordinates ($\varphi$: angular distance between different $\boldsymbol{a^G}$ orientations). Collinearity ($S$) was assessed by computing the dot product of normalized sensitivity vectors. (B) $\boldsymbol{\omega_{opt}}$ vectors were calculated using internal (head-bound) or external (earth-bound) angular velocity values (see Appendix). ε: angle of elevation of $\boldsymbol{\omega_{opt}}$ vectors relative to the (x,y) plane of the reference frame in which they were calculated. (C–D) $\boldsymbol{\omega_{opt}}$ vectors of two examples units, calculated using internal (left) or external (right) angular velocity, positioned at locations corresponding to orientations of $\boldsymbol{a^G}$ (in head coordinates) for which they were calculated, and color-coded according to their angle of elevation . (E–F) Collinearity of externally- (blue curve) and internally-referenced (red curve) sensitivity vectors vs. angular distance for the two units shown in *C* and *D*. (G) Difference between external and internal collinearity curves ($S$) for all units with significant rotational sensitivity (n = 66). Units with a strong external or internal tuning ($\Delta S > 0.5$ or $\Delta S < -0.5$ for $\varphi$ in the 80–100° range) are highlighted in blue (n = 7) and red (n = 5), respectively. The two units shown in *C* and *D* are highlighted in purple (unit 1, shown in *C*) and orange (unit 2, shown in *D*). (H) Direction of external sensitivity vectors for the seven units highlighted in blue in *G* (dark blue circles), and for 12 units with a weaker external tuning ($\Delta S > 0.1$ for in the 80–100° range, light blue circles), plotted on a pseudocylindrical projection. The excitatory direction of rotation of semi-circular canals is indicated as in *Figure 3F*.

The following figure supplement is available for figure 5:

**Figure supplement 1.** Rotational sensitivity map of a unit exhibiting no preferential tuning in an internal or external reference frame.

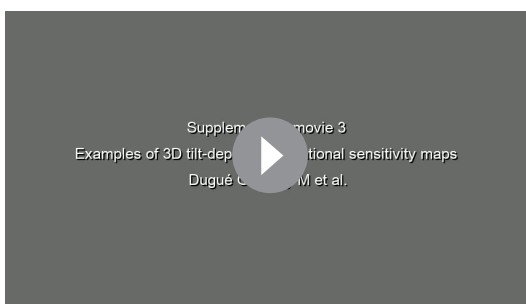

**Video 3.** Examples of 3D tilt-dependent rotational sensitivity maps. This movie shows the 3D tilt-dependent rotational sensitivity maps of the three example units shown in *Figure 4J*. To generate these plots, rotational sensitivity vectors were calculated (using either *internal* or *external* angular velocity values) for different head tilts (as explained in the Appendix) and plotted as arrowhead.

angular distance for internal but not external $\boldsymbol{\omega_{opt}}$ vectors in the left unit (*Figure 5E*), while it decreased for external but not internal $\boldsymbol{\omega_{opt}}$ vectors in the right unit (*Figure 5F*). Other units exhibited no clear preferred orientation for internal vs. external $\boldsymbol{\omega_{opt}}$ vectors, and collinearity curves that decayed similarly for the two reference frames (*Figure 5—figure supplement 1A, B*). When plotted for all units, the difference between collinearity curves of external and internal $\boldsymbol{\omega_{opt}}$ vectors revealed a continuum of properties (*Figure 5G*). Units exhibiting a stronger tuning toward an external or internal coding scheme were isolated by setting a cutoff (difference between external and internal collinearity > 0.5 or < −0.5 for angular distances ranging 80–100 °). All these units corresponded to putative Purkinje cells. The preferred rotation axes of external-coding cells did not appear to cluster around specific directions (*Figure 5H*).

## The sensitivity of caudal cerebellar units can be described as a head tilt-dependent tuning of linear responses to head angular velocity

As shown above, the mixed sensitivity of most units to rotational and gravitational information, captured by our model-free analysis, was confirmed by an empirical approach consisting in describing the units' rotational sensitivity as a linear tuning to a preferred rotation axis which depends on head tilt. This empirical approach showed that the units' sensitivity to angular velocity is highly dependent on head tilt, allowing in some cases the encoding of head rotations in a reference frame aligned with gravity.

We then compared the performance of different approaches to account for the observed firing rate modulations: 'local' (head-tilt dependent) linear models, and 'global' linear models (which simply assume a linear dependency of the firing rate on particular combinations of inertial parameters: rotational velocity, noted $\Omega$, angular acceleration, noted $\dot{\Omega}$, gravitational acceleration, noted $\mathrm{A}^{\mathrm{G}}$, and jerk of the gravitational acceleration, noted $\dot{\mathrm{A}}^{\mathrm{G}}$). These approaches were compared by computing their correlation with the observed instantaneous firing rate. As shown in *Figure 6A* (for the linear model based on $\Omega$, $\dot{\Omega}$ and $\mathrm{A}^{\mathrm{G}}$) and in *Figure 6—figure supplement 1* (for all other models), 'global' linear models always produced poorer predictions than our model-free approach. In contrast, the firing rate predictability calculated with our model-free approach was not significantly different from the coefficient of determination calculated from 'local' linear models used to generate tilt-dependent rotational sensitivity maps (p=0.24, paired Wilcoxon test, n = 86 units, *Figure 6B*). This highlights the presence of a nonlinearity in the way that caudal cerebellar units are tuned to head inertial parameters. This nonlinearity appears to be mainly due to a dependency to head tilt of the (linear) rotational sensitivity, since a series of 'local' linear fits of the unit's rotational sensitivity (head-tilt dependent $\boldsymbol{\omega_{opt}}$) collectively provide a description of the firing rate as performant as the model-free approach (*Figure 6B*).

## Discussion

Our findings reveal that the caudal cerebellar vermis hosts gravitationally polarized representations of head rotations in freely moving rats. We found that most putative Purkinje cell in the posterior cerebellum indeed exhibit receptive fields encoding head rotations about 3D axes anchored to the direction of gravity. This type of rotational sensitivity requires a complex re-encoding of head-centered sensory cues and might subserve downstream computations such as the encoding of head direction in the earth-horizontal plane.

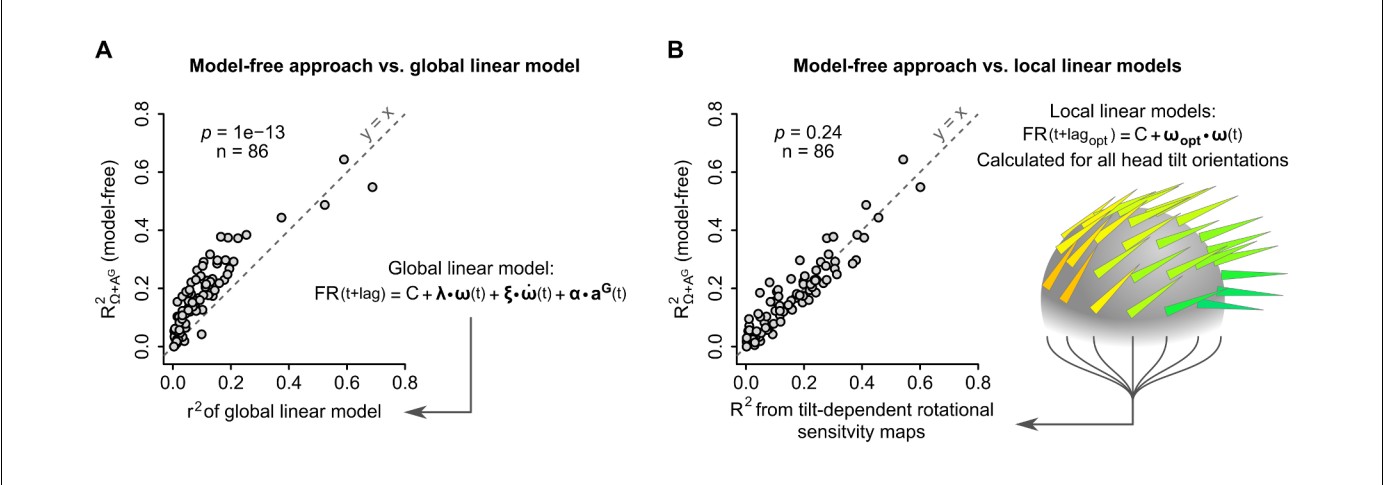

**Figure 6.** The model-free resampling approach is better at predicting firing rate than a 'global' linear model and is equivalent to a series of 'local' linear models calculated for different head tilts. (**A**) Firing rate predictability computed by the model-free resampling approach (see *Figure 1D–F* and Appendix) using angular velocity and gravitational acceleration ($R^2_{\Omega+\ A}{}^G$), plotted vs. the coefficient of determination ($r^2$) of a 'global' linear model describing the firing rate as a combination of angular velocity, angular acceleration, and gravitational acceleration (p=$1 \times 10^{-13}$, paired Wilcoxon test, see Appendix and *Figure 6—figure supplement 1*). Each point represents one unit (n = 86 units). (**B**) Firing rate predictability computed by the model-free resampling approach using angular velocity and gravitational acceleration ($R^2_{\Omega+\ A}{}^G$), plotted vs. the coefficient of determination of 'local' linear models describing the firing rate as a function of angular velocity and calculated for specific head tilts (p=0.24, paired Wilcoxon test). Each point represents one unit (n = 86 units). The arrowheads on the grey sphere symbolize 'local' optimal rotational sensitivity vectors ($\boldsymbol{\omega_{opt}}$) calculated for specific orientations of the gravity vector in the sensor frame (see *Figure 5A* and Appendix).

The following figure supplement is available for figure 6:

**Figure supplement 1.** Performance of the model-free resampling approach vs. global linear models.

## Inertial receptive fields in the caudal cerebellar vermis of freely moving animals

A key choice in our study was to approach the neuronal activity in the caudal cerebellum in freely moving animals, which express a large repertoire of spontaneous motor activity. The vestibular system has been mostly approached so far in head-fixed animals submitted to passive, externally applied, movements; however, studies in the vestibular and cerebellar nuclei during self-generated movements have provided strong indications that passive vs. active movements elicit different firing modulations in the same neurons (*Cullen et al., 2011*; *Medrea and Cullen, 2013*; *Cullen and Brooks, 2015*; *Brooks et al., 2015*). The use of spontaneously-generated movements yields specific constraints for the analysis: the movements have a high dimensionality (rotations and non-gravitational acceleration along 3 axes, head tilt, etc.), identical movements and head postures are not repeated over and over (as in passive conditions), and the repertoire of movements may not explore all the possible combinations: for example, we found that the data contained little purely translational acceleration (i.e. non gravitational acceleration which was not coupled to head rotations), which precluded the analysis of the influence of translations on the cells' firing. To deal with the high-dimensionality of the data, we used a two-step analysis: the first step involved a model-free approach, designed to identify (1) the fraction of instantaneous firing rate variability which could be explained by inertial parameters (head movement/tilt) and (2) which parameters best explained the observed firing rate fluctuations. Since the best prediction of firing rate for most units, particularly putative Purkinje cells, required to combine the knowledge of both head tilt (i.e. gravitational acceleration in the head reference frame) and angular velocity, we used a model where the firing rate is linearly tuned to a preferred rotation axis which depends on head tilt. We found that this model was as successful as the model-free approach to capture the units' sensitivity to inertial parameters.

We also identified a small set of units tuned to either head tilt or head angular velocity. Most of our granular layer units belonged to these categories, thus potentially reflecting the activity of

otolithic and semi-circular mossy fibers or Golgi cells directly driven by these fibers. A fraction of head tilt-selective units (n = 8) was identified as Purkinje cells, and might correspond to previously identified static roll-tilt Purkinje cells (*Marini et al., 1976*; *Yakhnitsa and Barmack, 2006*), or to tilt-selective Purkinje cells dynamically extracting head tilt information through multisensory integration (*Laurens et al., 2013a*; *Laurens et al., 2013b*).

## Cerebellar processing of inertial information

Our analysis is based on the units' instantaneous firing rate (rather than on spike times), as used previously in the cerebellum (e.g. *Ohtsuka and Noda, 1995*; *Pasalar et al., 2006*; *Medina and Lisberger, 2007*; *Yakusheva et al., 2007*). The rationale for this is that each units recorded may be viewed as a sample of a population with similar receptive fields, which shall converge on the same postsynaptic target; therefore the average behavior of a unit is similar to the instantaneous behavior of the population. The action of a single Purkinje cell on its postsynaptic target is likely to be small (*Bengtsson et al., 2011*) so that the intracellular potential of the target neuron is conditioned by the average activity of tens to a hundred Purkinje cells (*Person and Raman, 2011*). In the cerebellum, Purkinje cells are thought to be organized in microzones (for review, see *Apps and Hawkes, 2009*; *Dean et al., 2010*) which exhibit a narrow medio-lateral and large antero-posterior extension, which converge onto their downstream targets (*Sugihara et al., 2009*) and which share a similar climbing fiber teaching signal (as climbing fibers are thought to shape receptive fields). Indeed in the caudal vermis, the collaterals of climbing fibers follow the geometry of microzones (*Ruigrok, 2003*). We found that the instantaneous firing rate of many units exhibited strong fluctuations even during complete immobility. As a result, the instantaneous firing rate was only loosely connected to the rapid component of head inertial signals, consistent with a coding of information at the level of Purkinje cell populations (*Herzfeld et al., 2015*). In support of this interpretation, we found that neighboring putative Purkinje cells exhibited positive correlations of their firing rate and of the component of firing rate explained by inertial parameters (isolated with our model-free approach), indicating a similarity of the inertial receptive fields in cells likely belonging to the same microzone.

The majority of units in our sample exhibited tilt-dependent rotational tuning. Such units may intervene in the transformation of head-bound angular velocity signals (as sensed by the semi-circular canals) into externally referenced angular velocity signals, as observed in the forebrain (*Laurens et al., 2016*), or in azimuthal information within the head-direction system (*Taube et al., 2013*; *Finkelstein et al., 2016*; *Wilson et al., 2016*). Indeed, head tilt affects the correspondence between head-bound and earth-bound rotations (see *Video 4*). Therefore, cells tuned to rotations about an earth-bound axis are expected to exhibit a tilt-dependent sensitivity to head-bound rotations. How the activity of putative Purkinje cells is decoded in vestibular nuclear neurons shall largely depend on the (unknown) other inputs to these cells; however, as noted previously by *Yakusheva et al. (2008)*, sets of Purkinje cells (which are inhibitory neurons) encoding rotations relative to gravity (such as displayed in *Figure 5C*) could subtract the component of rotations that changes head orientation relative to gravity from semi-circular canal inputs in vestibular nuclear neurons; this operation would isolate an earth-horizontal component of head angular velocity suitable for the computation of azimuthal heading in head direction cells (*Finkelstein et al., 2016*).

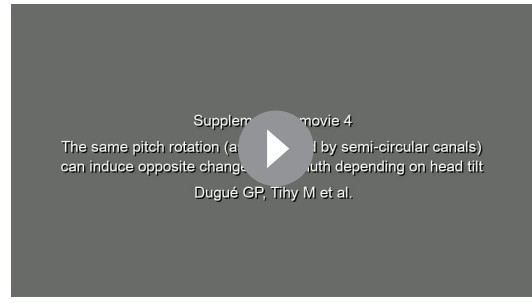

**Video 4.** Opposite changes of azimuth induced by the same pitch rotation (as measured by semi-circular canals) for two different head tilts. This movie illustrates how internally-referenced angular velocity signals do not necessarily map to externally-referenced angular velocity signals. Here, two identical pitch rotation sequences (as measured by the semi-circular canals) are executed from two different situations, in which the head is tilted either to the right or to the left. When the head is tilted right, an upward pitch rotation induces a change of azimuth (head angular direction in the earth-horizontal plane, represented in gray) toward the right; but when the head is tilted left, the same pitch rotation induces a change of azimuth in the opposite direction.

## Origin of the receptive fields in the caudal cerebellar vermis

The rotational sensitivity of caudal cerebellar units was maximal for positive lags relative to angular velocity, suggesting that it was driven by sensory cues rather than by motor commands. By comparing the firing rate predicted from sessions recorded in different conditions, we found no evidence for a crucial role of visual inputs, but instead found differences in the sensitivity to active vs. passive movements. This result is reminiscent of the lower sensitivity of certain vestibular nuclear neurons to active vs. passive movements (*Cullen and Roy, 2004*), and might reflect a similar mechanism of attenuation of self-generated inputs. Alternatively, the passive movements used in our study may have imperfectly sampled the inertial configurations encountered during active sessions, thereby complicating the comparison between the two conditions; indeed, self-generated movements often occurred at higher frequencies than our passive movements and may have induced smaller Purkinje cell modulations (*Yakusheva et al., 2008*). Further studies are required to identify the cause of differences of neuronal sensitivity to active vs. passive movements.

Deciphering how Purkinje cell receptive fields may be tuned to rotations about fixed directions relative to gravity is a complex topic. One challenge is to explain how rotational sensitivity is gated by gravity. Such an operation might be performed in the granular layer, by first computing an estimate of gravitational acceleration and then combining it with rotational (vestibular, visual or proprioceptive) informations. We found that the low-frequency (<2 Hz) component of acceleration during free movements mainly contains gravitational information. The granular layer of the caudal vermis contains a high amount of unipolar brush cells (excitatory interneurons intercalated between mossy fibers and granule cells; *Mugnaini et al., 2011*) which may smooth otolithic signals over hundreds of milliseconds (*van Dorp and De Zeeuw, 2014*; *Borges-Merjane and Trussell, 2015*; *Zampini et al., 2016*) and thus provide a proxy of gravitational signals to granule cells. Granule cells receiving convergent inputs (*Huang et al., 2013*) carrying gravitational and rotational information could then operate as coincidence detectors (*Chadderton et al., 2004*; *Chabrol et al., 2015*) and signal the occurrence of specific combinations of rotation and head tilt to Purkinje cells.

Purkinje cell receptive fields are shaped by the activity of climbing fibers which determine the sensitivity to subsets of granule cells (parallel fiber) afferents (e.g. *Dean et al., 2010*). Vestibular climbing fibers emanate from inferior olive neurons of the β-nucleus, which are controlled by dorsal Y-group and parasolitary nucleus afferents. These nuclei carry rotational and low-pass filtered otolithic signals but are also under the influence of vestibulo-cerebellar Purkinje cells (*Barmack and Yakhnitsa, 2000*; *Barmack, 2003*; *Wylie et al., 1994*). Therefore, the teaching signal sent to Purkinje cells by way of olivary neurons shall result from a complex interplay between external afferents and the action of Purkinje cells themselves, leading to the observed tilt-dependent rotational sensitivity.

## Conclusion

In conclusion, our work reveals that the dominant coding scheme of natural head kinematics in the caudal cerebellum is a tilt-dependent representation of head rotations. In a subset of cells, the mixed sensitivity to gravitational and rotational information was tuned in a way that allowed the encoding of head rotations in a gravity-centered reference frame. As emphasized previously (*Green and Angelaki, 2007*), the transformation of head-bound peripheral vestibular signals into an earth-bound representation of head kinematics involves complex nonlinear computations. In particular, the detailed mechanisms underlying the computation of gravitationally polarized rotational receptive fields remain to be elucidated. The presence of such receptive fields in the caudal cerebellar cortex, as shown here, as well as the key anatomical position of this structure immediately downstream of the vestibular organs and upstream of the vestibular nuclei, warrant a detailed examination of how the cerebellar cortical microcircuit might re-express head movements relative to gravity.

## Materials and methods

### Animals

Sixteen adult male Long-Evans rats (aged 3–4 months, 250–300 g at surgery, RRID:RGD_60991) were used in this study. Animals were housed individually in standard homecages maintained in

standard laboratory conditions (12 h day/night cycle, ≃21°C with free access to food and water). Experimental procedures were conducted in conformity with the institutional guidelines and in compliance with French national and European laws and policies. All procedures were approved by the 'Charles Darwin' Ethics Committee (project number 1334).

## Inertial measurement unit

Our inertial measurement unit (IMU) hosts a 9-axis digital inertial sensor containing a 3-axis accelerometer, 3-axis gyroscope and 3-axis magnetometer (MPU-9150, Invensense) soldered onto a custom printed circuit board (PCB, 9 × 7 mm) designed using EAGLE (CadSoft Computer GmbH). The sensor and additional passive components were soldered in-house using a reflow-soldering oven (FT-2000, CIF). Powering and communication with the IMU were performed using four wires that were soldered directly into wirepads present on the PCB. The IMU was connected to an I$^2$C interface (USB-8451, National Instruments) through a motorized commutator used for electrophysiological recordings (Tucker-Davis Technologies). Data were acquired at 250 Hz using a custom LabVIEW program. During experiments, the IMU was secured onto the rat's head using double row socket connectors (Mill-Max) as shown in *Figure 1A*.

## Tetrode drive

Our microdrive was designed based on the architecture proposed by *Anikeeva et al. (2011)*. The drive consisted of an M2.5 × 0.2 adjustment screw (F2D5ES15, Thorlabs) that was drilled along its axis (hole diameter: 1 mm) and placed inside an M2.5 × 0.2 adjustment knob (F2D5ESK1, Thorlabs). The knob and screw were inserted inside a custom aluminum housing machined in-house. The knob was machined such that it could be maintained in place while rotating using two interference pins. The screw was machined such that the housing prevented its rotation (thus rotation of the knob resulted in a translation of the screw, as in *Anikeeva et al., 2011*). The tetrode was threaded into a series of stainless steel tubings with increasing diameter and glued inside the screw such that at least 5 mm of tetrode protruded from the base of the microdrive with the screw in its upper position. The microdrive was designed to provide at least 3.5 mm of travel distance and weighted less than 3 g.

## Implantation surgery

Rats received an injection of the opioid analgesic buprenorphine (0.05 mg/kg s.c.) and were placed in a stereotaxic apparatus (model 942, David Kopf Instruments) under isoflurane anesthesia (at induction: 4%, 2 L/min; during surgery: 0.5–1.5%, 0.4–0.5 L/min). Body temperature was maintained between 37.4 and 37.6°C during the whole procedure using a regulated temperature controller coupled to a rectal probe (CMA 450, CMA). The scalp was shaved and wiped with povidone-iodine followed by 70% ethanol. Lidocaine (2%) was injected subcutaneously before incising the scalp. The skull was gently scraped with a scalpel blade and cleaned with a 3% hydrogen peroxide solution. Horizontal alignment of Bregma and Lambda was checked and burr holes were drilled (two over the frontal, four over the parietal and two over the interparietal bone plates) in order to insert skull screws (#0–80 × 3/32'' stainless steel screws, Plastics One). One of the screws was connected to a tungsten wire and used as a ground signal. A layer of self-curing dental adhesive (Super-Bond C&B, Sun Medical) was deposited over the skull except at the midline of the posterior half of the interparietal plate, where a craniotomy was drilled (interaural antero-posterior coordinate ranging −3.4 to −5.0 mm, that is, over the cerebellar lobule VI). After removing the dura, a single quartz-insulated tetrode (Thomas Recording) housed inside our custom microdrive and coated with DiI (Sigma) was lowered 4 mm below the surface of the brain, together with a 70 tungsten wire used as a reference. A drop of warm low-melting point agarose (1.5% in saline) was deposited around the tetrode, between the brain and the base of the microdrive to ensure brain mechanical stability. The base of the microdrive was then secured to the skull with dental cement (Pi-Ku-Plast HP 36, Bredent). A lightweight 3D-printed headstage hosting the IMU pin-connectors and a 16-channel electrode interface board (EIB, Neuralynx) was then placed over the microdrive and secured with dental cement. The four tetrode channels and the reference and ground wires were then connected to the EIB. The 3D-printed headstage contained a thin (3 mm) aluminium bar positioned horizontally along the interaural axis of the animal's head, and protruding 1.5 cm toward the right side. This bar was cemented in place with the rest of the implant and was used to immobilize the head during passive recording

sessions (see 'Recording protocol') using a custom clamping system. The skin ridges were sutured in front and at the rear of the implant and covered with antiseptic powder (Battle Hayward & Bower). A prophylactic injection of antibiotics was performed (gentamicin, 3 mg/kg, i.m.) and warm sterile saline was injected subcutaneously (2% of b.w.) to prevent dehydration. Rats were then allowed to recover under monitoring in their home cage placed over a heating pad. Gentamicin is known to produce vestibulotoxic effects when injected intratympanically or systemically over several days. Given the observations by *Oei et al. (2004)*, who only observed functional vestibular deficits after ten daily intramuscular injections of gentamicin at 50 mg/kg, it is highly unlikely that our single injection of gentamicin may have significantly affected the animal's vestibular organs.

## Electrophysiological recordings

Electrophysiological recordings were performed using a previously established protocol (*Gao et al., 2012*). Quartz insulated tetrodes (Thomas Recording) were gold-plated to reach an impedance of 150–200 kΩ at 1 kHz, mounted inside a custom microdrive (see 'Tetrode drive') and connected to an interface board (EIB-16, Neuralynx). Signals were referenced against a tungsten electrode positioned in the cerebellum 4 mm below the surface. The EIB was connected to a custom-made differential amplifier through a commercial headstage (Tucker Davis) and a motorized 32-channel commutator (Tucker Davis). Amplified signals were digitized at 30 kHz by a multifunction acquisition board (NI PCIe-6353, National Instruments) and acquired using a custom LabVIEW program.

Recordings were obtained from a volume of tissue comprised within the following interaural (IA) coordinates: 3.4–5 mm posterior to the IA point, ±5 mm around the midline and 2.5–5.4 mm above the IA point. This volume represents roughly less than 25% of the most caudal part of the vermis (lobules IX and X) according to *Paxinos and Watson, 2007*. Electrode tracks were examined using post hoc histology in eight rats. All tracks traversed lobules IX and X (*Figure 1—figure supplement 3F,G*).

## Recording protocol

After a 1-week post-operative recovery period, daily recording sessions were conducted as follows. The IMU and headstage were connected to the rat's head and the animal was placed inside a rectangular arena (120 × 60 cm). The animal was temporarily removed from the arena everytime the experimenter decided to lower the tetrode (by 1/8th of a turn, that is, 12.5 μm). The recordings were targeting zones of dense neuronal activity which typically correspond to Purkinje cell layers. Once units were obtained, the signal was controlled for stability and quality and electrophysiological and IMU recordings were started. A first block of 5 mm of free activity was obtained in the dark, followed by a second block of the same duration in the presence of light (active blocks). In some recordings, the head of the animal was immobilized for less than 10 s at the begining of the first block using a custom fixation system (see Implantation surgery), in order to provide enough signal to calculate gyroscope and accelerometer offsets.

The fixation system contained a small lightweight platform to which the body of the animal was strapped after immobilizing the head. To produce passive whole body movements, the platform was held by the experimenter and rotated about the roll, pitch and yaw axes, and about combinations of these axes. Two blocks were acquired in this condition (passive blocks), one in the dark and one in the light. A total of 17 units were recorded in both the active and passive condition (*Figure 2H*). Because several units could be isolated in the same site, these recordings corresponding to a total of 11 acquisition blocks in each condition. Because passive movements were produced by the experimenter, their kinematics differed from the rat's natural head kinematics. Nevertheless, the range of angular velocities was relatively similar in the passive vs. active condition (*Figure 2—figure supplement 1A*). Passive movements contained proportionally more low frequencies than active movements (*Figure 2—figure supplement 1B,C*), and the range of head orientations relative to gravity in the passive condition was slightly smaller (*Figure 2—figure supplement 1D*). Overall, recording sessions (in the active and passive condition) never lasted more than 3 hr in the same day.

## Data analysis

Full details of the analysis procedures and statistical tests are provided in the Appendix. Unless mentionned otherwise, p-values were obtained with an unpaired Wilcoxon test. All mean values are given with the standard deviation.

## Acknowledgements

We thank J Laurens, D Bennequin, A Afgoustidis and D Popa for helpful discussions on the project, and L Rondi-Reig and C Rochefort for initial discussions. We thank M Pasquet, B Barbour, R Proville, Y Cabirou and G Paresys for technical assistance. This work was supported by grants from France's Agence Nationale de la Recherche (ANR) to CL (ANR-12-BSV4-0027 and ANR-15-CE37-0001) and to BG (ANR-15-CE37-0007), and has received support under the program 'Investissements d'Avenir' launched by the French Government and implemented by the ANR, with the references ANR-10-LA-54 MEMO LIFE and ANR-11-IDEX-0001–02 PSL* Research University.

## Additional information

### Funding

| Funder | Grant reference number | Author |
| --- | --- | --- |
| Agence Nationale de la Recherche | ANR-15- CE37-0007 | Boris Gourévitch |
| Agence Nationale de la Recherche | ANR-12-BSV4-0027 | Clément Léna |
| Agence Nationale de la Recherche | ANR-10-LABX-54 MEMO LIFE | Clément Léna |
| Agence Nationale de la Recherche | ANR-11-IDEX-0001-02 PSL Research University | Clément Léna |
| Agence Nationale de la Recherche | ANR-15-CE37-0001 | Clément Léna |

The funders had no role in study design, data collection and interpretation, or the decision to submit the work for publication.

### Author contributions

GPD, Conceptualization, Data curation, Software, Formal analysis, Validation, Visualization, Methodology, Writing—original draft, Writing—review and editing; MT, Conceptualization, Data curation, Formal analysis, Investigation, Methodology, Writing—review and editing; BG, Conceptualization, Software, Formal analysis, Funding acquisition, Visualization, Writing—review and editing; CL, Conceptualization, Data curation, Software, Formal analysis, Supervision, Funding acquisition, Visualization, Methodology, Project administration, Writing—review and editing

### Author ORCIDs

Guillaume P Dugué, http://orcid.org/0000-0002-4106-6132
Matthieu Tihy, http://orcid.org/0000-0002-9314-4657
Boris Gourévitch, http://orcid.org/0000-0001-6742-8739
Clément Léna, http://orcid.org/0000-0002-1431-7717

### Ethics

Animal experimentation: Experimental procedures were conducted in strict conformity with the institutional guidelines and in compliance with French national and European laws and policies. All procedures were approved by the "Charles Darwin" Ethics Committee (project number 1334).

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

## Appendix 1

## Spike sorting and data post-processing

To isolate spikes, continuous wide-band extracellular recordings were first filtered offline with a Butterworth 1 kHz high-pass filter. Spikes were then extracted by thresholding the filtered trace and the main parameters of their waveform were extracted (width and amplitude on the four channels). The data were hand-clustered by polygon-cutting in two-dimensional projections of the parameter space using Xclust (Matt Wilson, MIT). The quality of clustering was evaluated by inspecting the units' auto-correlograms, as in *Gao et al. (2012)*.

A time series representing the instantaneous firing rate (*FR*) was obtained by taking the inverse of the local interspike interval (ISI) every millisecond. The resulting step-like function was convolved with a 10 ms Gaussian to soften the transitions between successive ISIs. This smoothing had no deleterious incidence on the correlations calculated between *FR* and inertial parameters given the fact that head movements occurred with significantly longer time scales (>50 ms, see *Figure 1—figure supplement 1C*). The instantaneous firing rate time series was then downsampled at 200 Hz to match the times of inertial measurements.

## Classification of recorded units

To classify the recorded units, we used criteria derived from *Van Dijck et al. (2013)* (*Figure 1—figure supplement 3E*). Most units exhibited a spiking entropy between 6 and 8, consistent with a Golgi or Purkinje cell profile. Among them, units with low firing rate (cut-off set at 12 Hz) were classified as putative Golgi cells (n = 4) and units with a higher firing rate were classified as putative Purkinje cells (n = 78), consistent with observations in the posterior vermis of monkeys. Because tetrodes pick many units at the same time, and because of the high level of activity in the Purkinje cell layer of awake rats, we could distinguish complex spikes from simple spikes only in a few recordings (*Figure 1—figure supplement 3H*). Putative mossy fibers (n = 4) were distinguished based on their characteristic low entropy (*Van Dijck et al., 2013*) and separated from other units using a cut-off of 6.3.

Spike entropy characterizes the regularity of firing (the higher the less regular). It was computed as previously described (*Van Dijck et al., 2013*) from the histogram $p(\mathrm{ISI_i})$ of the natural logarithm of interspike intervals (in milliseconds) taken with a binwidth of 0.02 and smoothed with a gaussian kernel with a standard deviation of one-sixth of the histogram's standard deviation. The entropy was then defined as (*n* being the number of bins):

$$\mathrm{Entropy} = \sum_{i=1}^{n} p(\mathrm{ISI_i}) \cdot \log_2 p(\mathrm{ISI_i}) \tag{1}$$

## Analysis of acceleration signals

### Rationale of the analysis

The inertial parameters measured by the sensor and the animal's vestibular organs can be decomposed in:

- An angular velocity signal ($\Omega$, sensed by semi-circular canals and by the sensor's gyroscopes).
- An acceleration signal (A, sensed by otolith organs and the sensor's accelerometers), which can be decomposed in gravitational acceleration ($A^G$), calculated using an orientation filter (as explained below), and non-gravitational acceleration ($A^{nG}$). $A^{nG}$ contains purely

translational acceleration ($A^T$), as well as linear tangential acceleration ($A^{LT}$) and centripetal acceleration ($A^C$), both resulting from rotations.

Hence:

$$A = A^G + A^{nG} \qquad (2)$$

$$A^{nG} = A^T + A^{LT} + A^C \qquad (3)$$

$\Omega$, $A^G$ and $A^T$ are experienced identically at any point on a solid. Because the sensor was fixed to the skull via a connector embedded in dental cement and because vestibular organs are skull-anchored structures, $\Omega$, $A^G$ and $A^T$ were experienced identically by the sensor and the animal's vestibular organs.

This section details the calculations for estimating $A^G$ and for assessing the contribution of head rotations to the production of non-gravitational acceleration signals. Because the quantification of $A^{LT}$ and $A^C$ requires to know the distance between the sensor and the axes about which rotations are executed, and because this distance can not be precisely determined, the calculations detailed below are only meant to provide an order of magnitude for these rotation-dependent components of acceleration, and to assess their global contribution to $A^{nG}$. In other words, the goal of these calculations was not to quantify $A^{LT}$ and $A^C$ precisely, but to assess whether $A^{nG}$ was likely to be dominated by pure translational acceleration signals or by acceleration signals produced by head rotations.

## Estimation of gravitational acceleration

An estimate of gravitational acceleration was calculated using Sebastian Madgwick's IMU sensor fusion algorithm (also known as 'orientation filter', *Madgwick et al., 2011*). For every time point, the algorithm computes a quaternion representation of a 3D rotation in the Earth frame that brings the sensor from an initial orientation to its current orientation (*Figure 1— figure supplement 1D1*). The converse rotation can be used to keep track of the orientation of the gravity vector in the sensor frame (*Figure 1—figure supplement 1D2*), and thus to calculate the gravitational components of acceleration along the 3 axes of the sensor (*Figure 1B*). Because gyroscopes provide measurements of angular velocity with a certain amount of noise, the temporal integration of gyroscope signals leads to accumulating error in the estimation of orientation. The purpose of the algorithm is to use acceleration signals (which, on long timescales, reflect mostly the effect of gravity) to compute the direction of this error at every time step, and to compensate for it by subtracting the magnitude of the noise of gyroscope measurement ($\beta$) along that direction (in our conditions, $\beta$ was in the order of 0.1 °/s, as per component specifications and according to recordings performed from an immobile IMU). This procedure ensures that the estimated orientation remains properly aligned with gravity, thereby providing accurate head tilt (attitude) information, but does not guarantee accurate heading information. In other words, the algorithm does not provide accurate information on the *absolute* orientation of the sensor in the Earth frame, but only on its orientation relative to gravity.

## Linear tangential and centripetal acceleration resulting from head rotations

Linear tangential acceleration ($A^{LT}$) is produced in the tangential direction of movement, perpendicular to the axis of rotation, during a change of angular velocity (angular acceleration). Centripetal acceleration ($A^C$) is exerted in the radial direction toward the center of rotation. The magnitude of these accelerations depends on the distance $d$ between the sensor and the axes about which head rotations are executed. Because $d$ can

not be precisely determined, the following calculations are only meant to provide an order of magnitude for $A^{LT}$ and $A^C$. For these calculations, we made the following assumptions:

- That most head rotations are pitch (up/down) and yaw (left/right) rotations (as shown in **Figure 1—figure supplement 1B**). Because roll rotations are executed about and axis that passes near the sensor, the contribution of this type of rotation to $A^{LT}$ and $A^C$ can be considered as minimal.
- That pitch/yaw head rotations are executed about axes of rotation that pass through the animal's neck, and that the distance $d$ between the sensor and these rotations axes is in the order of several cm (as defined during the implantation surgery).

For the calculation, we thus discarded $\omega_x$ values measured by the sensor, which concentrate most of the roll information, and assumed that pitch/yaw rotations move the head around a resting posture (pitched 35° nose down, as shown in **Blanks and Torigoe, 1989**). In this default orientation, the sensor is tilted as shown in **Figure 1—figure supplement 2C1** and thus captures the yaw component of rotations (here defined as left-right rotations) along its $x$ and $z$ axes. The yaw component of angular velocity produces a linear tangential acceleration along the $y$ axis of the sensor (**Figure 1—figure supplement 2C1**) that can be calculated as follows:

$$a_y^{lt} = d \cdot \frac{\delta\left(\frac{\omega_z}{\cos(\theta)}\right)}{\delta t} \tag{4}$$

where $\theta$ is the tilt angle of the head relative to an upright orientation (**Figure 1—figure supplement 2C1**), $d$ is the distance between the sensor and the center of rotation and $a_y^{lt}$ is the linear tangential acceleration along $y$ in m·s$^{-2}$. Pitch rotations produce linear tangential accelerations along the $x$ and $z$ axes of the sensor (**Figure 1—figure supplement 2C1**) which can be calculated as follows:

$$a_x^{lt} = d \cdot \frac{\delta\omega_y}{\delta t} \cdot \sin(\theta) \tag{5}$$

$$a_z^{lt} = -d \cdot \frac{\delta\omega_y}{\delta t} \cdot \cos(\theta) \tag{6}$$

The norm of the linear tangential acceleration vector $\boldsymbol{a^{lt}}$ produced by pitch/yaw rotations (**Figure 1—figure supplement 2D**) was then calculated as:

$$|\boldsymbol{a^{lt}}| = \sqrt{a_x^{lt^2} + a_y^{lt^2} + a_z^{lt^2}} \tag{7}$$

The norm of the centripetal acceleration vector $\boldsymbol{a^c}$ (pointing toward the center of rotation) resulting from pitch/yaw rotations (**Figure 1—figure supplement 2D**) was calculated as:

$$|\boldsymbol{a^c}| = d \cdot \left(\omega_{yaw}^2 + \omega_y^2\right) \tag{8}$$

# Estimation of firing rate modulations that are due to head movements

## Principle of the analysis

The goal was to implement a model-free, data-driven method to test the intensity and robustness of the link between a set of external variables (here the inertial parameters of

head movements) and the fluctuations of the instantaneous firing rate of recorded units. The core computation (detailed below) employs a jackknife resampling technique that calculates firing rate estimates based on the repeated occurrence of similar combinations of values in the external variables considered.

The square of the Pearson correlation coefficient between the estimated and observed firing rates is taken as a measure of the influence of these external variables on a unit's activity. It accounts for the amount of firing rate variance that can be explained by these variables, and thus can be considered as a measure of how well these variables can predict firing modulations (firing *predictability*).

In practice, other factors may contribute to firing rate fluctuations (such as a high discharge irregularity at rest or other types of external variables) and may reduce the observed correlation. To test the consistency of our estimation method (i.e. the *robustness* of the link between the external variables and firing rate fluctuations), we used a cross-validation approach consisting in calculating the Pearson correlation coefficient between independent firing rate estimates obtained using non-overlapping portions of the same recording.

## Steps of the analysis

For the sake of clarity, the different steps of the analysis are listed below. These different steps, which are described in detail in the next paragraphs, are performed independently for each cell and provide a metric of how well inertial parameters can explain fluctuations of the instantaneous firing rate (firing rate *predictability*), and of how reliable firing rate estimation can be with these parameters (*robustness* of the prediction).

1. Calculation of the instantaneous firing rate (FR, see 'Spike sorting and data post-processing'). FR and raw inertial signals are transformed into **time series** with the same sampling interval.
2. Estimation of the **gravitational component of acceleration** ($A^G$) using an orientation filter algorithm (see 'Estimation of gravitational acceleration').
3. Each category of signal ($\Omega$, A, $A^G$, $A^{nG}$) is **normalized** (see 'Firing rate predictability').
4. **Optimal time delays** (lags) between FR and each category of signal are estimated by cross-correlation (see 'Firing rate predictability').
5. For each point in the recording, a **resampling procedure** is used to calculate the expected firing rate at that point given firing rate values observed for points with similar combinations of inertial parameters (see 'Firing rate predictability'). The operation is repeated for each time point to obtain an **estimated firing rate time series**. For this step, combinations of inertial parameters can be chosen arbitrarily.
6. The square of the Pearson correlation coefficient between the estimated firing rate time series and FR is taken as a measure of **firing rate predictability** using a specific combination of inertial parameters (see 'Firing rate predictability').
7. The **robustness of the prediction** is assessed by computing independent estimated firing rate time series using non-overlapping portions of the recording (see 'Robustness of the prediction'). The Pearson correlation coefficient between these independent estimates is taken as a measure of the robustness of the estimation.

## Firing rate predictability

A set of $n$ time-dependent external variables $(v_1, v_2, ..., v_n)$, here representing the inertial parameters of head movements, and recorded within a time interval $[0, T]$, can be described as a time-dependent vector $V(t) = (v_1(t), v_2(t), ..., v_n(t))$, with $t \in [0, T]$. At each time $t_0$ in $[0, T]$, an estimate of the instantaneous firing rate can be computed as the average of firing rate values observed at times during which $V$ is similar to $V(t_0)$. In practice, the predicted firing rate is calculated using only a set of points where the Euclidean distance between $V$ and $V(t_0)$ is the smallest, discarding the points immediately preceding or following $t_0$. The procedure is detailed below.

Because the variables in $V$ may span very different values (accelerations, measured in $g$, are typically in the $[-2, 2]$ range while angular velocity values, measured in °/s, are typically in the

$[-200, 200]$ range), we first rescaled the variables (grouped by type, either acceleration or angular velocity) by dividing them by their group SD, yielding normalized versions of the variables $V^{norm} = (v_1^{norm}, v_2^{norm}, ..., v_n^{norm})$. For example, the normalization of angular velocity values was performed as follows:

$$SD_\Omega = SD([\omega_x(t), \omega_y(t), \omega_z(t)]_{t \in [0,T]})$$

$$\omega_x^{norm}(t) = \frac{\omega_x(t)}{SD_\Omega}; \omega_y^{norm}(t) = \frac{\omega_y(t)}{SD_\Omega}; \omega_z^{norm}(t) = \frac{\omega_z(t)}{SD_\Omega}$$

Before grouping the variables into the vector $V^{norm}$, we had to take into account the fact that there was potentially a lag between these variables and their effect on the firing rate. For each category of variable, the optimal lag $\tau$ was taken as the temporal delay at which the highest cross-correlation was observed between the instantaneous firing rate and variables of that category. For angular velocity:

$$\tau_{opt}^\Omega \text{ maximizes } f(\tau) = max(\{|\omega_x * FR(\tau)|, |\omega_y * FR(\tau)|, |\omega_z * FR(\tau)|\}_{\forall t})$$

where $\omega_x * FR$ is the correlation function of $\omega_x$ and FR. The normalization and time-shifting steps above yield the following vector of variables:

$$\begin{aligned} V^{norm}(t) = (&\omega_x^{norm}(t - \tau_{opt}^\Omega), \omega_y^{norm}(t - \tau_{opt}^\Omega), \omega_z^{norm}(t - \tau_{opt}^\Omega), \\ &a_x^{norm}(t - \tau_{opt}^a), a_y^{norm}(t - \tau_{opt}^a), a_z^{norm}(t - \tau_{opt}^a), ...) \end{aligned} \tag{9}$$

where $(\omega_x, \omega_y, \omega_z)$ and $(a_x, a_y, a_z)$ represent respectively angular velocity and net acceleration values along the sensor's axes, $\tau_{opt}^\Omega$ is the optimal lag for angular velocity and $\tau_{opt}^a$ the optimal lag for net acceleration. Note that $V^{norm}$ can incorporate other variables calculated offline, such as gravitational and non-gravitational acceleration (see 'Analysis of acceleration signals').

For each $t_0$ in $[0, T]$, we then calculated the Euclidean distance between $V^{norm}(t_0)$ and all other values of $V^{norm}$ encountered in $[0, T]$:

$$\forall t \in [0, T], \ d(t_0, t) = dist(V^{norm}(t_0), V^{norm}(t)) \tag{10}$$

We then selected the $N$ smallest values of $d\{d(t_0, t_1), ..., d(t_0, t_N)\}$ ($N = 200, 500$ or $1000$), excluding time points that were too close to $t_0$ (exclusion window = $\pm 0.5$ s in our calculations, which corresponds to 100 points before and after $t_0$, given our sampling frequency of 200 Hz). The firing rate estimate at $t_0$ was then calculated as the average of firing rate values observed at the time points corresponding to these $N$ values of $d$:

$$\rho_V(t_0) = \underset{t \in \{t_1, ..., t_N\}}{mean}\{FR(t)\} \tag{11}$$

The process was then repeated for every time point in $[0, T]$ to obtain a complete firing rate estimate $\rho_V$ over the whole recording. This firing rate estimate was then compared to a version of the instantaneous firing rate computed using the local average ISI value (taking into account ISIs occurring before and after each ISI). This smoothed version of the instantaneous firing rate ($FR_{smooth}$) was calculated as follows. For each time point $t_0$ falling inside the $j^{th}$ ISI, the value of $FR_{smooth}$ was given by:

$$\mathrm{FR}_{\mathrm{smooth}}(t_0) = \frac{2n+1}{\sum_{k=j-n}^{j+n} \mathrm{ISI}_k} \tag{12}$$

where $n$ is the number of ISIs before and after the $j^{th}$ ISI (in our calculation, $n = 2$). $\mathrm{FR}_{\mathrm{smooth}}$ was calculated at 1 kHz and then convolved with a 10 ms Gaussian kernel (as in 'Spike sorting and data post-processing'). The square of the Pearson correlation coefficient ($R^2$) between $\rho_V$ and $\mathrm{FR}_{\mathrm{smooth}}$ was taken as a measure of the fraction of firing rate modulation that could be explained by $V$ (firing rate *predictability*). We found empirically that comparing $\rho_V$ with $\mathrm{FR}_{\mathrm{smooth}}$ and not the instantaneous firing rate calculated from the inverse of the ISI ($n = 1$, as defined in Spike sorting and data post-processing) yielded about 40% greater $R^2$ values. The interpretation of these higher values is that $\mathrm{FR}_{\mathrm{smooth}}$ softens the fluctuations of the ISI (*Figure 1—figure supplement 3D*), smoothing them down to a timescale closer to the one of movements (*Figure 1—figure supplement 1C*).

Note that the resampling procedure can be performed using arbitrary sets of external variables, and thus the influence of different sets of variables on firing rate can be tested and compared (*Figure 1E*). We used the following notation to differentiate firing rate estimates calculated using differents combinations of inertial parameters: $\rho_\Omega$ when considering angular velocity only, $\rho_{A^G}$ when considering gravitational acceleration only, $\rho_{\Omega,A^G}$ when considering both, etc.

## Robustness of the prediction

To test the robustness of the link between a set of external variables $V$ and firing rate modulations, we computed independent firing rate estimates using non-overlapping portions of the same recordings. In practice, we split the interval $[0, T]$ of each recording into two separate ensembles by selecting alternatively 10 second-long time intervals:

$$\mathrm{set}_1 = \{[0,10[, [20,30[, [40,50[, ...\} \text{ and } \mathrm{set}_2 = \{[10,20[, [30,40[, [50,60[, ...\}$$

We then calculated two firing rate estimates ($\rho_{V_1}$ and $\rho_{V_2}$) using time points (and the corresponding values of $V$) from either one or the other dataset (i.e. $\rho_{V_1}$, resp. $\rho_{V_2}$, was calculated for all time points of the recording using only data from $\mathrm{set}_1$, resp. $\mathrm{set}_2$). The Pearson correlation coefficient ($R$) between $\rho_{V_1}$ and $\rho_{V_2}$ was then calculated and taken as a measure of the robustness of the firing rate estimation procedure. The distribution of $R$ values for all units is shown in *Figure 1F*. To calculate these values, the best combination of inertial parameters (i.e. providing the greatest firing predictability) was selected on a cell-by-cell basis. A null distribution of $R$ values was obtained by randomly shuffling the ISIs of the units (*Figure 1F*). Values in this distribution are the average $R$ calculated over 10 iterations (each with an independent ISI shuffling step).

## Comparing the robustness of firing rate predictions for different recording conditions

A similar cross-validation strategy was used to evaluate the consistency of the link between $V$ and firing rate modulations when the experimental conditions were changed (e.g. when recordings were performed in the light or in the dark, or when head movements were self-generated or passively experienced). The logic was the following: if the link between $V$ and firing rate modulations remains unchanged between two blocks recorded in different conditions ($\mathrm{block}_1$ and $\mathrm{block}_2$), then the similarity (i.e. the Pearson correlation coefficient) between two independent firing rate estimates in $\mathrm{block}_1$ ($\rho_{V_1}^{\mathrm{block}_1}$ and $\rho_{V_2}^{\mathrm{block}_1}$) should be equivalent to the similarity between the firing rate estimate in $\mathrm{block}_1$ ($\rho_V^{\mathrm{block}_1}$) and a firing rate estimate in $\mathrm{block}_1$ computed with the dataset from $\mathrm{block}_2$ ($\rho_V^{\mathrm{block}_2 \rightarrow 1}$).

For the comparison of blocks recorded in the light or dark (*Figure 2E,F*), we first noticed that the combination of inertial parameters yielding the best firing rate estimate for each cell ($\mathbf{V}_{best}$) was always the same for the two conditions. We then compared the similarity between independent firing rate estimates for the light block ($\rho_{V_{best,1}}^{\text{light}}$ and $\rho_{V_{best,2}}^{\text{light}}$) to the similarity between the firing rate estimate for the light block ($\rho_{V_{best}}^{\text{light}}$) and a firing rate estimate for the light block computed using the dark block dataset ($\rho_{V_{best}}^{\text{dark}\rightarrow\text{light}}$).

For the comparison of blocks recorded in the presence of active or passive movements (*Figure 2G,H*), the best predictive model for each cell ($\mathbf{V}_{best}$) was first identified, irrespective of the condition (active or passive). Using this model, the similarity between independent firing rate estimates for the passive block ($\rho_{V_{best,1}}^{\text{passive}}$ and $\rho_{V_{best,2}}^{\text{passive}}$) was compared to the similarity between the firing rate estimate for the passive block ($\rho_{V_{best}}^{\text{passive}}$) and a firing rate estimate for the passive block computed using the active block dataset ($\rho_{V_{best}}^{\text{active}\rightarrow\text{passive}}$).

## Receptive fields of rotation-selective units

### Inertio-temporal receptive fields

These plots (*Figure 3B* and *Figure 4A,D*) were obtained by calculating the average timecourse of the instantaneous firing rate around specific values of angular velocity ($\omega_x$, $\omega_y$ or $\omega_z$). The variable considered (e.g. $\omega_x$) was first divided into equal intervals (binwidth = 5 °/s). Every point in the recording was assigned to its corresponding interval and the instantaneous firing rate trace within a given time window around that point (lag of ±0.5 s) was saved. To build inertio-temporal receptive fields, firing rate traces were averaged for each interval. The resulting firing rate values were then color coded and represented as a function of the variable (center value of each interval) and the lag.

### 3D plots of rotational sensitivity

These plots (see for example *Figure 3C* and *Video 2*) were obtained by calculating the average instantaneous firing rate for specific combinations of the three components of angular velocity ($\omega_x$, $\omega_y$ and $\omega_z$) at a given lag. These variables were first divided into equal intervals (binwidth = 25 °/s). Every point in the recording was assigned to a specific combination of $\omega_x$, $\omega_y$ and $\omega_z$ intervals and the instantaneous firing rate at the optimal lag (see 'Linear fits') relative to that point was saved. Firing rate values belonging to the same combination of intervals were averaged, color coded and represented in 3D as a function of angular velocity values (center values of the intervals).

### Linear fits

The instantaneous firing rate of rotation-selective units at time $t + lag$ (*lag* defined by the user) was described as a linear combination of angular velocity values at time $t$ (see *Figure 3D*):

$$\text{FR}(t + lag) = C + \alpha \cdot \omega_x(t) + \beta \cdot \omega_y(t) + \gamma \cdot \omega_z(t) \tag{13}$$

The coefficients of the fit define a *sensitivity vector* $\boldsymbol{\omega}_{lag} = (\alpha, \beta, \gamma)$ whose direction represents the unit's 3D preferred axis and direction of rotation, and whose norm $|\boldsymbol{\omega}_{lag}| = \sqrt{\alpha^2 + \beta^2 + \gamma^2}$ represents the unit's response gain (i.e. sensitivity, in Hz/(°/s) or $°^{-1}$) at the lag considered. The above linear fit formula is equivalent to:

$$FR(t + lag) = C + \boldsymbol{\omega_{lag}} \bullet \boldsymbol{\omega}(t) \tag{14}$$

where $\boldsymbol{\omega_{lag}} \bullet \boldsymbol{\omega}(t)$ is the dot product of $\boldsymbol{\omega_{lag}}$ and the instantaneous angular velocity vector $\boldsymbol{\omega}(t) = (\omega_x(t), \omega_y(t), \omega_z(t))$. The *optimal sensitivity vector* ($\boldsymbol{\omega_{opt}}$) is obtained at the lag yielding the largest sensitivity (*optimal lag*). The direction of $\boldsymbol{\omega_{opt}}$ indicates the unit's preferred 3D axis and direction of rotation. Its norm $|\boldsymbol{\omega_{opt}}|$ represents the unit's sensitivity along its preferred rotation axis. To assess the significativity of rotational sensitivity, an average $|\boldsymbol{\omega_{lag}}|$ vs. lag curve was generated for each unit using shuffled spike trains (constructed from the original ISI distribution; $n = 100$ iterations; lag range = $[-0.5, 0.5\,\text{s}]$). The average norm of $\boldsymbol{\omega_{lag}} + 3.5 \times SD$ calculated from this curve was taken as the significativity threshold. The sensitivity of a unit was considered significant if the $[-0.5, 0.5\,\text{s}]$ lag range contained $|\boldsymbol{\omega_{opt}}|$ values greater than the significativity threshold.

## Tilt-dependent rate maps

Tilt-dependent rate maps indicate normalized average firing rate values calculated for different head orientations (**Figure 3H**). Head orientation relative to gravity (head tilt) was identified by the direction of the gravity vector $\boldsymbol{a^G} = (a_x^G, a_y^G, a_z^G)$ in the sensor frame (see **Video 1**). During rotations causing a reorientation of the head relative to gravity, $\boldsymbol{a^G}$ describes a trajectory contained over a sphere of radius 1 $g$ in the sensor frame. To examine how the instantaneous firing rate was modulated by head-tilt, we selected 500 points evenly distributed over that sphere (**Figure 3G**). We then calculated the average instantaneous firing rate for all time points during which $\boldsymbol{a^G}$ fell within 20° of each of these points (discarding points for which less than 200 time points were found in the recording). The resulting average firing rate values were normalized as follows:

$$FR_{norm,i} = \frac{FR_i - \underset{i \in \{1,...,N\}}{\text{mean}}\{FR_i\}}{\underset{i \in \{1,...,N\}}{\text{mean}}\{FR_i\}} \tag{15}$$

The sphere was tiled with polygons centered around these points using a Delaunay triangulation and the resulting spherical map was projected in 2D using the Lambert azimutal equal-area projection (the center of the map corresponding to an upright head orientation, that is, with the head-vertical axis aligned with $\boldsymbol{a^G}$). Finally the polygons were filled with a color indicating the corresponding normalized average firing rate value (**Figure 3H**). The non-parametric coefficient of variation of these values (**Figure 3I**) was given by:

$$CV_{npar} = \frac{Q(0.84) - Q(0.16)}{\underset{i \in \{1,...,N\}}{\text{median}}\{FR_{norm,i}\}} \tag{16}$$

where $Q(0.84)$ and $Q(0.16)$ are the quantiles of the normalized average firing rate distribution corresponding to the 0.84 and 0.16 probabilities (the probabilities corresponding to $\pm 1SD$ in a gaussian distribution).

## Stability index

To quantify the effect of head tilt on the direction of preferred rotational sensitivity vectors over a defined lag range, we first computed sensitivity vectors ($\boldsymbol{\omega_{lag}} = (\alpha_{lag}, \beta_{lag}, \gamma_{lag})$) for negative and positive values of $a_x^G$ or $a_y^G$ for lag values comprised between $-0.5$ and $0.5$. Sensitivity vectors were normalized as follows:

$$\omega_{lag}^{norm} = \frac{\omega_{lag}}{\sqrt{\sum\limits_{lag=-0.5}^{0.5} \alpha_{lag}^2 + \beta_{lag}^2 + \gamma_{lag}^2}} \tag{17}$$

The stability index $\sigma$ (**Figure 4G,H**) was calculated as follows:

$$\sigma_{a_x^G} = \sum\limits_{lag=-0.5}^{0.5} \omega_{lag,a_x^G>0}^{norm} \bullet \omega_{lag,a_x^G<0}^{norm} \tag{18}$$

$$\sigma_{a_y^G} = \sum\limits_{lag=-0.5}^{0.5} \omega_{lag,a_y^G>0}^{norm} \bullet \omega_{lag,a_y^G<0}^{norm} \tag{19}$$

The above normalization ensures that $\sigma$ is bounded between $-1$ and $1$ (1 indicating strictly identical directions and $-1$ indicating strictly opposite directions over the lag range).

## Calculation of angular velocity in a reference frame aligned with gravity

The orientation filter (see 'Estimation of gravitational acceleration') allowed us to compute the direction of the gravity vector $a^G$ in the sensor frame. An *external* right-handed reference frame was defined using the following constraints: a z axis aligned with $a^G$, and an x axis within the plane defined by $a^G$ and the head's naso-occipital axis and pointing toward the animal's nose. The components of the instantaneous angular velocity vector, originally expressed in the sensor frame, were calculated in this external reference frame and used to compute externally-referenced optimal rotational sensitivity vectors using a linear model (as in 'Linear fits').

## Tilt-dependent rotational sensitivity maps

To compute head tilt-dependent optimal rotational sensitivity vectors, we used a strategy similar to the one employed for tilt-dependent rate maps. We first selected 500 points evenly distributed over a sphere and calculated $\omega_{opt}$ vectors (see 'Linear fits') using only observations during which directions of $a^G$ fell within 20° of each of these points (discarding points for which less than 500 observations in the recording were found). These vectors were computed using the internal or external (see 'Calculation of angular velocity in a reference frame aligned with gravity') components of angular velocity signals and represented on a sphere at the coordinates corresponding to the direction of $a^G$ for which they were calculated (**Figure 5C,D** and **Video 3**).

To quantify the performance of this approach, we identified for each time point the corresponding head tilt and thus the corresponding $\omega_{opt}$ vector. This allowed us to generate a firing rate prediction for each time point, and to correlate this prediction with the observed firing rate. The square of the Pearson correlation coefficient ($R^2$) obtained by this method was then compared to the $R^2$ calculated using the model-free approach (see 'Estimation of firing rate modulations that are due to head movements' and **Figure 6B**).

## Collinearity between rotational sensitivity vectors

The collinearity of pairs of rotational sensitivity vectors $(\boldsymbol{\omega}_{opt,i}, \boldsymbol{\omega}_{opt,j})$, calculated for two different orientations of the gravity vector $(a_i^G, a_j^G)$ in the sensor frame, was assessed by calculating the dot product:

$$S_{i,j} = \widehat{\boldsymbol{\omega}}_{opt,i} \bullet \widehat{\boldsymbol{\omega}}_{opt,j} \tag{20}$$

To compute collinearity profiles (average collinearity vs. angular distance $\varphi$ between $\boldsymbol{a}^G$ orientations), $S_{i,j}$ was calculated for all pairs $(i,j)$ with $i\,\&\,j \in \{1,...,N\}$ and $i \neq j$, where $N$ is the number of different $\boldsymbol{a}^G$ orientations for which sensitivity vectors were calculated (among the 500 points selected over the sphere, see 'Tilt-dependent rotational sensitivity maps'). These values were then averaged as a function of $\varphi$ and plotted for bins of $\pi/16$ (**Figure 5E,F**).

## Global linear models

To compare the performance of our model-free approach (see 'Estimation of firing rate modulations that are due to head movements') with simple linear regressions, we designed 'global' linear models. These models are called 'global' because they are calculated using the whole recording, and not a subset defined for example by specific head orientations relative to gravity (like in tilt-dependent rotational sensitivity maps, described in section 'Tilt-dependent rotational sensitivity maps'). The linear model described in section 'Linear fits' is a particular case of such model. In these models, the instantaneous firing rate (FR) is described as a linear combination of different categories of inertial parameters, with a lag defined by the user. Optimal lags were calculated by testing lag values comprised between –0.5 and 0.5 s. For each model, the optimal lag was defined as the one that maximized the model's coefficient of determination. The following models were used:

$$\mathrm{FR}(t + \mathrm{lag}) = C + \boldsymbol{\lambda} \bullet \boldsymbol{\omega}(t) \tag{21}$$

$$\mathrm{FR}(t + \mathrm{lag}) = C + \boldsymbol{\lambda} \bullet \boldsymbol{\omega}(t) + \boldsymbol{\xi} \bullet \dot{\boldsymbol{\omega}}(t) \tag{22}$$

$$\mathrm{FR}(t + \mathrm{lag}) = C + \boldsymbol{\alpha} \bullet \boldsymbol{a}^G(t) \tag{23}$$

$$\mathrm{FR}(t + \mathrm{lag}) = C + \boldsymbol{\alpha} \bullet \boldsymbol{a}^G(t) + \boldsymbol{\beta} \bullet \dot{\boldsymbol{a}}^G(t) \tag{24}$$

$$\mathrm{FR}(t + \mathrm{lag}) = C + \boldsymbol{\lambda} \bullet \boldsymbol{\omega}(t) + \boldsymbol{\alpha} \bullet \boldsymbol{a}^G(t) \tag{25}$$

$$\mathrm{FR}(t + \mathrm{lag}) = C + \boldsymbol{\lambda} \bullet \boldsymbol{\omega}(t) + \boldsymbol{\xi} \bullet \dot{\boldsymbol{\omega}}(t) + \boldsymbol{\alpha} \bullet \boldsymbol{a}^G(t) \tag{26}$$

$$\mathrm{FR}(t + \mathrm{lag}) = C + \boldsymbol{\lambda} \bullet \boldsymbol{\omega}(t) + \boldsymbol{\xi} \bullet \dot{\boldsymbol{\omega}}(t) + \boldsymbol{\alpha} \bullet \boldsymbol{a}^G(t) + \boldsymbol{\beta} \bullet \dot{\boldsymbol{a}}^G(t) \tag{27}$$

where $\boldsymbol{\omega}(t)$, $\dot{\boldsymbol{\omega}}(t)$, $\boldsymbol{a}^G(t)$ and $\dot{\boldsymbol{a}}^G(t)$ are 3D vectors representing the instantaneous angular velocity, angular acceleration, gravitational acceleration and jerk (time derivative) of the gravitational acceleration, respectively, and $\boldsymbol{\lambda}$, $\boldsymbol{\xi}$, $\boldsymbol{\alpha}$ and $\boldsymbol{\beta}$ are 3D vectors representing the corresponding coefficients of the fit. The coefficients of determination of these linear models (at their optimal lag, defined for each unit) were compared to the firing rate predictability calculated by our model-free approach (see 'Estimation of firing rate modulations that are

due to head movements') using angular velocity and gravitational information (see *Figure 6A* and *Figure 6—figure supplement 1*).

