## [Decision Letter]

Thank you for submitting your article "Cerebellar re-encoding of self-generated head movements" for consideration by *eLife*. Your article has been reviewed by three peer reviewers, and the evaluation has been overseen by a Reviewing Editor and a Senior Editor. The following individuals involved in review of your submission have agreed to reveal their identity: Sergio Carmona (Reviewer #3).

The reviewers have discussed the reviews with one another and the Reviewing Editor has drafted this decision to help you prepare a revised submission.

Summary:

This manuscript describes an elegant set of experiments examining the reference frames and neural computations used to encode self-motion from vestibular cues. The responses of cerebellar neurons in the caudal cerebellar vermis to natural vestibular stimuli in freely moving animals, and the recording in freely moving animals is a great strength of this approach. The results indicate that a majority of cerebellar Purkinje cells encode head rotations in a head tilt-dependent manner, and a subset of cells encode head rotations in a gravity/earth-centered reference frame, rather than the head-centered reference frame of the vestibular periphery. These findings about how natural head movements are represented in the cerebellum suggest a possible mechanism for computing head direction for spatial orientation, and thus should be of broad interest.

Essential revisions:

1) The data analysis is complex, and needs to be more clearly described in the main text and justified. In particular, the model free analysis should be more clearly described and justified.

The reviewers struggled to determine whether the parameters (weights and time delays) were determined from the average of instantaneous firing rate of each cell, average from all cells, or changed from case to case. More generally, there was concern about the unstated assumptions, and potential noise and accumulation of small errors associated with the multiple layers of analysis (data smoothing, effectively data binning, multidimensional fitting algorithms across both time and parameters whose amplitudes are normalized, calculating correlation coefficients and then taking correlations of correlation coefficients).

The description around Figure 1, which assesses the robustness of the analysis, is confusing. For 80% of the units, the Pearson correlation coefficient (R) between independent estimates was greater than 0.6. How was this value of 0.6 selected? What would the Pearson coefficients look like for shuffled data, or some type of null distribution? What about the other 20% of the neurons? If this method is one to confirm the veracity of the analysis, doesn't this imply that for 20% of the neurons the method does not work or that the unit was not the same one throughout the recording? What influenced the decision to keep such units (particularly the 8 or so units with low or no correlation)?

Does the model-free approach do significantly better than more traditional approaches? For smooth pursuit eye movements, the instantaneous firing rate of Purkinje cells does a great job of representing the kinematics as a linear sum of velocity, acceleration and the position of the eye. Does this more standard approach fail for the current data?

The value of looking at sensitivity vs. lag (Figure 3) is not clear.

In Figure 4, wouldn't the degree of nose up/down tilt effect the yaw sensitivity? The significance of Figure 4 should be clarified or those panels not included.

2) There is currently considerable debate as to whether one can use the firing of cerebellar neurons to identify them as a specific cell type. The authors should use additional criteria to confirm cell type identity, or back off on their claims about which cell types were recorded.

3) The "passive whole body movements condition" is very poorly described. How well matched are the vestibular stimuli that the animals experience in the passive and active condition? If they are not similar in the spatial or temporal dynamics, it seems this might explain the poorer correlations in Figure 2. Could the smaller amount of data for the passive condition be contributing to the discrepancy?

4) Once sub-divided, some of the groups have Ns of less than 5. Can the number of recordings increased to generate more confidence in these divisions and perhaps unravel additional ones which might have been missed?

5) The analysis appears to require knowledge of the distance d, between the IMU and the point around which yaw and pitch movements are made. It is not clear whether the absolute distance is important, and if so, how it is determined. If (as it appears) the non-gravitational linear acceleration is not important for understanding the behavior of the neurons, this should be made clearer (subsection “Cerebellar units exhibit a mixed sensitivity to head angular velocity and gravitational acceleration”).

---

## [Author Response]

Essential revisions:

*1) The data analysis is complex, and needs to be more clearly described in the main text and justified. In particular, the model free analysis should be more clearly described and justified.*

We are grateful to the reviewers for pointing out the lack of clarity of the main text. We have modified our manuscript to provide a clearer description and justification of each step of the analysis.

The data analysis is mostly directed at assessing the link between the firing rate and the inertial parameters of head movements. In brief, we first look at how much of the instantaneous firing rate fluctuations can be explained by the inertial parameters and at how consistent this link is across conditions (these questions are addressed using a model-free approach in Figure 1 and Figure 2). We then explore this link using linear models (Figure 3 to 6) and show that, except for a small subset of units which show a rather simple tuning to angular velocity or head tilt (Figure 3), the sensitivity of most units can be modeled as a head-tilt-varying linear dependency to head rotations (i.e. with a “preferred” rotation that depends on head orientation relative to gravity; Figure 4 and Figure 5). We show that this model performs as well as the model-free approach (but, as we now show in the revised version of the manuscript, gives better results than simple linear models of the instantaneous firing rate; see Figure 6 and Figure 6—figure supplement 1).

The model-free approach was used to determine how much (and which set of) inertial parameters influence the instantaneous firing rate. The choice of a model-free approach as an initial method for analyzing the data was motivated by the fact that non-linearities in the responses of primary vestibular afferents have been reported during naturalistic movements (Schneider et al., 2015), and non-linearities in the cells’ sensitivity to head inertial signal are expected if head movements are re-encoded in an external reference frame (Green and Angelaki, 2007). This approach makes the assumption that similar values of inertial parameters yield (with an optional delay) similar instantaneous firing rate values; it does not hypothesize any a priori model. The approach yields an estimate of the average instantaneous firing rate driven by a given set of inertial parameters; this estimate can also be viewed as the instantaneous output of a population of neurons sharing similar sensitivity. An important caveat for the use of this approach is its sensitivity to the sampling of the parameter space by the behaving animal during the session, which determines whether the “neighbors” in the parameter space used to average the firing rate are close enough to build a consistent estimate. We found in general a good consistency (robustness) of the estimation as shown by the correlation coefficient between independent firing rate estimates (Figure 1), indicating that in most cases, the sampling of the parameter space was sufficient. Finally, this approach was used to compare the units’ sensitivity across conditions (e.g. light and dark) without making any assumptions on this sensitivity: firing rate estimates for a given session may be generated using either the same or another session as a “training” session (i.e. neighbors in the inertial parameter space may be searched in the same or in another session), and compared (Figure 2). A noticeable point is the computational cost of this approach which typically require several hours of CPU per unit.

The alternative method used to analyze the data is composed by multiple “local” linear models, assuming a linear tuning to a preferred rotation which depends on the head orientation (Figure 5). This method seems as good as the method-free approach to capture the fraction of firing rate fluctuation due to the inertial parameters considered. This is now more clearly stated in the manuscript and highlighted with Figure 6 (formerly Figure 5—figure supplement 1).

We made substantial efforts to justify and describe more thoroughly the model-free approach in the revised manuscript and in the revised Appendix section and to better justify the steps of analysis.

*The reviewers struggled to determine whether the parameters (weights and time delays) were determined from the average of instantaneous firing rate of each cell, average from all cells, or changed from case to case. More generally, there was concern about the unstated assumptions, and potential noise and accumulation of small errors associated with the multiple layers of analysis (data smoothing, effectively data binning, multidimensional fitting algorithms across both time and parameters whose amplitudes are normalized, calculating correlation coefficients and then taking correlations of correlation coefficients).*

We apologize for any lack of clarity of the description of the procedures. We added a summary of all the analysis steps for the model-free approach in the revised Appendix section. We summarize here the key points.

There is no use of weights in the analysis; on the other hand, we use lags to take into account that fact that the firing rate may not be instantly linked to the inertial parameters, but that there may be a delay between the two. The lags are computed for each cell in each recording block. In the case of the model-free approach, the lag is taken as the time delay that maximizes the cross-correlation between the firing rate and each category of inertial parameter (for reason of computational cost, we did not attempt to optimize this choice by seeking the lags giving the largest correlation between the estimated and observed instantaneous firing rates). In the case of linear models, the lag is chosen to yield the largest coefficients (hence, the largest sensitivity) in a simple linear regression predicting the firing rate from angular velocity signals (although this simple linear model does not perform as well as the combination of linear models calculated for different head tilts, shown in Figure 5, we felt it would not make much sense to use different lags for different head tilts). Average sensitivity vs. lag curves are shown in Figure 3.

We summarize below the main steps of the analysis.

In the model-free analysis, the calculations are:

1) Downsampling of inertial signal: some inertial data were acquired at 200 Hz and the other were thus down-sampled from 250 Hz to 200 Hz; we felt this 200 Hz frequency acceptable in view that 1) the time constant of head movements is longer than 50 ms (see our response to point 1 in the minor revisions), 2) the transmission delay between the sensor and the acquisition system exhibited a small jitter of a 1–2 milliseconds (Pasquet et al., 2016), therefore limiting the temporal precision of the measures under 5 ms.

2) Computing an instantaneous firing probability. We calculated a time series representing an estimate of the instantaneous firing probability as a function of time. Each time point of the time series was defined as the inverse of the ISI during which this point fell.; the resulting step-like function was then convolved with a 10 ms Gaussian distribution to soften the transitions between successive ISIs (see minor point 1 below for more justification on this procedure).

3) Normalizing the different types of inertial parameters (angular velocity, acceleration). This is needed to allow the identification of neighboring points within the high dimensional space of inertial parameters. Indeed, the model-free resampling procedure requires, for a given (“current”) time point, to find similar combinations of inertial parameters at other time points in the recording (taking into account time lags, as discussed above) and to average the corresponding firing rate values. Similar combination of inertial parameters are identified by the fact that their Euclidean distance to the “current” combination is minimal (see Appendix section). Given the fact that angular velocity and acceleration signals span ranges of values that differ by two orders of magnitude ( ± 200 °/s and ± 2 g, respectively), the calculation of this Euclidean distance only makes sense if both types of signals are normalized.

4) The optimal time delay (lag) used for the resampling procedure (described above) was calculated for each cell using a simple cross-correlation analysis (see above). In other words, we calculated an optimal lag for each category of signal (angular velocity, total acceleration, gravitational acceleration, non-gravitational acceleration) for each cell in each recording block.

5) We then run the resampling procedure. For a given time point in the recording, this step consists in identifying the nearest neighbors in the parameters space (time points with a combination of inertial parameters similar to the point considered) and calculating an estimated (predicted) firing rate value for this point by averaging the corresponding instantaneous firing rate values. This step is repeated for all points in the recording to obtain a time series representing the estimated firing rate.

6) We then compare this estimated firing rate with the instantaneous firing rate; we found that the correlation was low, largely because the instantaneous firing rate exhibits fast fluctuations due to irregularities in the units’ discharge (CV of ISI ~1); we therefore chose to compare the estimated firing rate with a smoothed version of the observed instantaneous firing rate, obtained by averaging a few ISIs around each time point (as explained Appendix section). This smoothed instantaneous firing rate provides a more consistent measure of the instantaneous firing probability (e.g. more stable instantaneous firing rate values at rest) at the cost of a slower time course (which is however adaptive: the instantaneous firing rate evolves faster when all ISIs are short than when all ISIs are long). Indeed, we found empirically that linear fitting and model-free approaches were generally better at predicting an instantaneous firing rate computed by averaging a few ISIs (which is therefore smoother but is still modulated at the time scales of variations of the inertial parameters); the number of ISIs used (2 on each side, see Appendix section) was chosen empirically as the one giving the best match between the estimated (by the model-free approach) and the smoothed instantaneous firing rate.

For the tilt-dependent rotational sensitivity maps:

1) We compute an optimal lag from a global linear model (firing rate as a linear combination of the angular velocity components, see Appendix section)

2) The data are binned by head-tilt.

3) Within each head-tilt bin, we compute a “local” (i.e. head-tilt dependent) linear model linking the firing rate to the angular velocity (using the optimal lag defined above), which provides a head-tilt dependent preferred rotation (see Appendix section). No normalization is involved.

Other reviewers concerns:

1) Data binning was used to calculate inertio-temporal receptive fields (Figure 3, Figure 4), 3D plots of rotational sensitivity (Figure 3), tilt-dependent rate maps (Figure 3) and tilt-dependent rotational sensitivity maps (Figure 5). No binning takes place in the model-free approach. Inertio-temporal receptive fields and tilt-dependent rate maps are used for data inspection only (and illustration purposes); the choice of bin is a trade off between resolution and noise. In the case of rotational sensitivity maps, we found that using less than 500 time points in the recording yielded poor estimates of the unit’s preferred rotation; the choice of the “bins” of head tilt (in which to compute the preferred rotation) was a trade off between this requirement and obtaining this preferred rotations for a large fraction of the different head tilts explored by the animal (i.e. for smaller bins, only a few head positions would be visited frequently enough to provide the 500 values needed to compute the preferred rotation).

2) “multidimensional fitting algorithms across both time and parameters whose amplitudes are normalized”: this may refer to the model-free approach (which is not exactly a fitting procedure since it simply calculates the average instantaneous firing rate for comparable combinations of inertial signals, and then compares it to the original instantaneous firing rate). Justification for the normalization is provided above. The fitting procedures in the manuscript are otherwise linear regression but they do not operate over normalized variables.

3) “calculating correlation coefficients and then taking correlations of correlation coefficients”. Correlations coefficients are used to evaluate the performance of the model-free approach. The correlation between the estimated and the observed firing rates is a measure of how well a set of inertial parameters can explain the observed firing rate fluctuations (they provide a measure of firing rate predictability, Figure 1). We did not specifically calculate correlations of correlation coefficients, but the reviewers’ comment may refer to correlation between independent firing rate estimates (these estimates are rather averages than correlations); these computations are used to measure of how robust the estimation is (Figure 1). The robustness of firing rate estimates were compared (using a Wilcoxon test) when an experimental condition was changed, e.g. when recordings were performed in the presence or absence of light (Figure 2) or when movements were actively or passively generated (Figure 2). The rationale for this analysis is the following: the model-free approach provides an estimate of the fraction of firing rate variance determined by the inertial parameters; for a given set of inertial signals we can use different recording sessions to perform the prediction of the firing rate. We can then compare the two firing rates predicted from the two sessions by computing their correlation coefficients; if this coefficient is 0, it means that the link between the firing rate and movements in the two sessions (at least as assessed by our method) is entirely uncorrelated. However, two predictions obtained from the same condition might also differ to a certain degree if the overall contribution of inertial parameters to firing rate fluctuations is small (e.g. in the presence of other factors influencing the firing rate or because of differences in sampling of the parameter space); therefore, to assess whether a low correlation coefficient between two firing rates predicted from two different conditions is due to a change of sensitivity between the two conditions, we compared this correlation coefficient to the correlation between two firing rates estimates predicted from the same condition: if both are low, this indicates that there is globally a poor predictability of the cell, while a lower correlation across conditions versus within condition will indicate a change of sensitivity (which appears to be the case in the active vs. passive condition; see our response to point 3 of the essential revisions).

We hope that our efforts for providing a clearer explanation and justification of our analysis in the revised manuscript and Appendix section methods will dispel the reviewers’ concerns about the analysis framework.

*The description around Figure 1, which assesses the robustness of the analysis, is confusing. For 80% of the units, the Pearson correlation coefficient (R) between independent estimates was greater than 0.6. How was this value of 0.6 selected? What would the Pearson coefficients look like for shuffled data, or some type of null distribution? What about the other 20% of the neurons? If this method is one to confirm the veracity of the analysis, doesn't this imply that for 20% of the neurons the method does not work or that the unit was not the same one throughout the recording? What influenced the decision to keep such units (particularly the 8 or so units with low or no correlation)?*

We thank the reviewers for pointing out this confusing presentation. Figure 1 shows the distribution of correlation coefficients between independent firing rate estimates (the measure of the robustness of the estimation, as explained above in our response to point 1.1) calculated for each cell using the model-free approach. The value of 0.6 indicated in the text was given for descriptive purpose, as the point was to simply illustrate the fact that most units had a large correlation coefficient. This has been removed.

To clarify the significance of the values of this correlation coefficient (noted R), and as suggested by the reviewer, we calculated the distribution of R values obtained from shuffled spike trains (“null distribution”): for each cell, an average value of R was calculated over 10 iterations, each performed using an independent spike shuffling step. The resulting null distribution is shown in the revised Figure 1 in red. It is significantly different from the original distribution (p = 2.2 x 10^[16]^, Wilcoxon test), with a mean value of 0.14 ± 0.10 (vs. 0.65 ± 0.23 for the original distribution). [NB: the original distribution (i.e. in gray) is slightly different in this revised panel as we had failed in the previous version to properly select R values corresponding to the optimal combination of inertial parameters (i.e. the one that gives the greatest predictability). This was corrected in the current version of the plot.]

As noted by the reviewers, a few neurons indeed display R values that lie within the limits of the null distribution. This is unlikely to result from unstable recording or from successive appearance of two units in the same recording: the calculation of R is in principle insensitive to these situations since R is computed from firing rate estimates computed using short non-overlapping portions of the recording that are interleaved throughout the whole recording block (see Appendix section). Hence these two estimates should be equally sensitive to changes in the recording across time, and their correlation should not be affected by these changes (we verified explicitly that cells with small R values did not show large Gini index on the ISIs, which is a measure of the stability of the units’ firing rate along time). On the other hand, small R values are found for cells with little of no tuning to head movements or with limited sampling (animal less active). We kept these units for the subsequent analyses, which largely involved linear models, as the linear fit may be less sensitive to reduced sampling of head movements than the model-free approach (i.e. fewer points may suffice to extract a linear trend), and may still be able to pick up correlations between the firing rate and head movements. The significativity of sensitivity of the cells provided by the linear models was independently assessed (see the bootstrap analysis for the significativity of linear models in the Appendix section); a few of the cells with low R value indeed exhibited a low, but significant sensitivity to head rotations.

The part of the main text describing this part of the results was rewritten to take into account the above comments, and the calculation of the null distribution is detailed in the revised version of the Appendix section).

*Does the model-free approach do significantly better than more traditional approaches? For smooth pursuit eye movements, the instantaneous firing rate of Purkinje cells does a great job of representing the kinematics as a linear sum of velocity, acceleration and the position of the eye. Does this more standard approach fail for the current data?*

The use of a model-free approach needed to be better justified. We thank the reviewer for point this out.

As explained in the response to point 1 of the essential revisions, the original motivation for using the model-free approach is that it may pick up nonlinearities that could be missed by linear models. These nonlinearities could be for example a nonlinear transfer function of primary vestibular afferents in the amplitude range covered by naturalistic movements (as shown by Schneider et al., 2015), or nonlinearities associated to a change of reference frame as observed for acceleration in primate cerebellum (i.e. coordinate transformations, as explained by Green and Angelaki, 2007).

As suggested by the reviewers, we now provide a comparison of the performance of the model-free approach with the one of “global” linear models. The terms of the models were different combinations of angular velocity, angular acceleration, gravitational acceleration (head tilt) and the jerk (time derivative) of gravitational acceleration. When comparing the coefficient of determination of these linear regressions with the R2 of the model-free approach computed with angular velocity and gravitational acceleration, we found that the model-free approach was always better at explaining firing rate fluctuations than these models. These results are detailed in a new paragraph in the main text of the revised manuscript (last section of the results) and shown in a new Figure 6 and a new Figure 6—figure supplement 1. The calculation of global linear models is detailed in the Appendix section.

Strikingly, the series of “local” linear models of rotational sensitivity (used to estimate the preferred rotation as a function of head tilt in “tilt-dependent rotational sensitivity maps”) performed as well as the model-free approach (Figure 6, formerly in Figure 5—figure supplement 1). This suggests that, for each head tilt, the cells are linearly tuned to the head’s angular velocity, but this linear dependency depends on head tilt in a nonlinear way. The similarity of the performance of the model-free approach and of this model (that assumes a linear dependency to head rotation changing with head-tilt) suggests that this later model is close to the best prediction possible in most cells in our conditions.

*The value of looking at sensitivity vs. lag (Figure 3) is not clear.*

This plot is related to the exploration of the cells’ sensitivity to angular velocity using a linear model (as shown in Figure 3 and explained in the Appendix section). The coefficients of the linear fit provide a measure of a cell’s sensitivity (in Hz/(deg/s) for a given lag chosen by the user. By running the linear model for each cell using different lags, we identified “optimal lags” (i.e. that maximize the calculated sensitivity) on a cell-by-cell basis. By showing how the calculated sensitivity of units is strongly modulated by the choice of the lag used, Figure 3 illustrates the importance of identifying optimal lags and shows differences in the average sensitivity and optimal lag value for the 3 categories of units distinguished using the model-free approach (gravitational, rotational and mixed, identified in Figure 3). [NB: we show the average tuning curve in Figure 3, but the optimal lag was computed for each cell].

To better explain the importance of this graph, we modified the corresponding text in the revised manuscript.

*In Figure 4, wouldn't the degree of nose up/down tilt effect the yaw sensitivity? The significance of Figure 4 should be clarified or those panels not included.*

The overall aim of Figure 4 is to illustrate how the sensitivity of the firing rate to head angular velocity is affected by head tilt, taking into account the temporal dimension. We were therefore facing the challenge of representing firing rate as a (non-linear) function of multidimensional data: angular velocity along the three axes, gravitational acceleration along the three axes and temporal lag. We focused on characteristic examples, in which the sensitivity of a cell to the yaw rotation axis is modulated by pitch tilt, reducing the analysis to 4 dimensions: yaw velocity, pitch tilt, firing rate and temporal lag. Despite this reduction in the dimensions of the analysis, the information can only be displayed using multiple graphs.

Figure 4 are inertio-temporal receptive fields (as in Figure 3) that show how the firing rate varies in time around specific yaw velocity values, and that were calculated for positive and negative values of the gravitational acceleration along *x* (reflecting the pitch-tilt angle); we therefore plotted the firing rate (color value) as a function of the time delay (abscissa) and yaw velocity at zero lag (ordinate) by grouping respectively all nose-up (right graph in panels A and D) and nose-down (left graph in panels A and D) orientations. This plot shows qualitatively different inertio-temporal receptive fields for the head-up and head-down cases (which we feel are useful to show), but indeed the pooling of all nose-up and nose-down orientations prevents a proper assessment of the effect of different degrees of pitch tilt on the sensitivity of these example units. To examine how the sensitivity to yaw velocity changes with the degree of pitch tilt, we plotted the firing rate (color value) as a function of pitch tilt (abscissa) and yaw velocity (ordinate) at the optimal lag in panels C and F; these plots thus do not show the time dependency of panels A and D. Panel C indeed shows that the firing rate of this example cell is much increased (respectively decreased) by leftward (respectively rightward) rotations when the nose is up while much fewer modulation takes place when the nose is down. Panel F instead shows that the firing rate of another cell is increased by leftward rotations when the nose is down while it is decreased by similar rotations when the nose is up; the progressive change of sensitivity (i.e. firing rate modulation along the ordinate axis) as a function of head-tilt (abscissae axis) is indeed visible on these panels.

Panels B and E are quantifications of panels A and D. They represent the slope of the firing rate vs. yaw velocity relation for discrete lag values (i.e. taken along vertical lines of the 2D histograms in panels A and D). They allow to visually confirm the change of sensitivity to yaw velocity for positive vs. negative pitch tilt angles. We edited panels B and E to include a measure a statistical significance using a bootstrap method (the gray areas represent the limits of the mean sensitivity slope ± 2SD calculated using shuffled spike trains).

We hope that these explanations will convince the reviewer that these plots are complementary and help provide the reader with a view of the multiple dependencies of the firing rate.

*2) There is currently considerable debate as to whether one can use the firing of cerebellar neurons to identify them as a specific cell type. The authors should use additional criteria to confirm cell type identity, or back off on their claims about which cell types were recorded.*

We thank the reviewer for raising this point. The activity was recorded presumably in the Purkinje cell layer which is noticeable for its dense activity. Indeed, the firing features of most units recorded fit with the classically-observed characteristics of Purkinje cells (Figure 1—figure supplement 3; van Dijck et al., 2013). However, despite our best efforts, complex spikes proved very hard to distinguish from simple spikes except for a few exceptions (Figure 1—figure supplement 3). Therefore, the compelling argument to prove that most of our units are Purkinje cells is lacking. The text has been amended accordingly by reminding the reader that units’ identification is tentative (the identity of the units is always preceded by the word “putative”). We still single out the few units which exhibit firing characteristics more consistent with those of Golgi and mossy fibers, since these units tend to show simpler receptive field (A^G^-selective and Ω-selective) as displayed in Figure 3, and thus contrast with the main population of putative Purkinje cells recorded in our study.

*3) The "passive whole body movements condition" is very poorly described. How well matched are the vestibular stimuli that the animals experience in the passive and active condition? If they are not similar in the spatial or temporal dynamics, it seems this might explain the poorer correlations in Figure 2. Could the smaller amount of data for the passive condition be contributing to the discrepancy?*

We provide a more detailed explanation of how passive whole-body movements were produced in the revised manuscript (paragraph “Recording protocol” of the Material and methods). Because these movements were produced by the experimenter, who held the platform to which animals were fixed, their detailed statistics necessarily (despite his best efforts) differed from the movements produced spontaneously by the animals in the freely moving condition.

To quantify these differences, we calculated the statistics of active vs. passive movements in recording blocks that corresponded to the 17 units that were recorded in the two conditions (Figure 2). The results are shown in a new supplementary figure (Figure 2—figure supplement 1). Although passive movements were produced by the experimenter, the ranges of angular velocities were in fact quite similar (the only significant difference was found for the range of roll velocity values as shown in Figure 2—figure supplement 1). Passive movements contained proportionally more low frequencies than active movements (Figure 2—figure supplement 1), and the range of head orientations relative to gravity (head tilts) was slightly smaller in the passive condition (Figure 2—figure supplement 1). Therefore, movements in the passive vs. active conditions were not perfectly matched, but did not differ drastically (perfectly matching passive movements with active movements would require a robotic device able for example to replay the range of active movements; this technically challenging approach was not possible in the present study). Overall, the passive conditions appeared to explore a subset of the inertial configurations explored in the active condition; this is the reason why we only attempted to predict (and compare) the firing rate of the passive condition using the passive and active sessions (Figure 2): there would be too many inertial configurations in the active condition without a correspondence in the passive session to successfully predict the active firing rate by sampling the passive session. However, we can still not exclude that the difference between active and passive sessions revealed by our model-free approach results from differences between the movements; this is now more clearly stated in the text.

*4) Once sub-divided, some of the groups have Ns of less than 5. Can the number of recordings increased to generate more confidence in these divisions and perhaps unravel additional ones which might have been missed?*

The main result of this manuscript is the demonstration of the existence of a large group of cells showing a head-tilt-varying sensitivity to head rotation. Few cells show discharge properties which are atypical for Purkinje cells and have been classified as putative Golgi cells and putative mossy fibers (see point 4). These cells could indeed be removed from the presentation, but we feel their inclusion indeed enriches the (small) set of cells showing simple sensitivities (to rotation or to gravity) shown in the Figure 3; moreover, it allows us to introduce several analysis procedures on simple cases (inertio-temporal receptive fields, preferred rotational sensitivity vector, spherical presentation of head tilt) which are then re-used in cells exhibiting more complex receptive fields (Figure 4 and Figure 5). Overall several relatively small groups, “A^G^–selective” units, “Ω–selective” units, “internal” units, “external” units size (n = 12, 6, 5, 7) appear as extreme cases of somewhat continuous properties.

We apologize for not being able to add new data, but we have used in this manuscript all the recordings available and generating new data is a task which would require much more time than allotted for this revision.

*5) The analysis appears to require knowledge of the distance d, between the IMU and the point around which yaw and pitch movements are made. It is not clear whether the absolute distance is important, and if so, how it is determined. If (as it appears) the non-gravitational linear acceleration is not important for understanding the behavior of the neurons, this should be made clearer (subsection “Cerebellar units exhibit a mixed sensitivity to head angular velocity and gravitational acceleration”).*

Because the sensor is attached on the animal’s skull, the angular velocity and the gravitational component of acceleration are experienced identically at the level of the sensor and of the animal’s vestibular organs and do not require to know the distance *d* between the sensor and the rotation axes. As shown by our model-free approach (Figure 1) and confirmed by our tilt-dependent rotational sensitivity maps (see Figure 6 and our response to point 1.3 of the essential revisions), these are the two key parameters that modulate the units’ responses; the non gravitational acceleration does not seem to play an important role (Figure 1); therefore changing the value of *d* would not change our understanding of the units’ behavior.

In Figure 1—figure supplement 2, we calculated an estimation of the magnitude of two types of non-gravitational acceleration signals that are coupled to rotations (linear tangential and centripetal acceleration) and thus depend on the value of *d*. The purpose was only to estimate their order of magnitude and their contribution to non-gravitational acceleration signals. Because *d* cannot be precisely determined, we assumed that its value is in the cm range, which is a realistic assumption given the placement of the sensor on the animal’s head. Our calculation shows that a large fraction of non-gravitational acceleration most probably arises from linear tangential acceleration rather than centripetal or purely translational acceleration (Figure 1—figure supplement 2). Therefore, the repertoire of behavior spontaneously expressed by the rats in our condition may not be most adequate for studying the dependency of firing rate on translational acceleration.

We edited the text in the revised Appendix section to provide clearer explanations.